# Dive into the Scene: Breaking the Perceptual Bottleneck in Vision-Language Decision Making via Focus Plan Generation

Boyuan Xiao [1]  Bohong Chen [1]  Yumeng Li [1]  Ji Feng [2]  Yao-Xiang Ding [1 3]  Kun Zhou [1]

## Abstract

In embodied vision-language decision making tasks such as robotic manipulation and navigation, Vision-Language and Vision-Language-Action Models (VLMs & VLAs) are powerful tools with different benefits: VLMs are better at long-term planning, while VLAs are better at reactive control. However, their performance is limited by the same perceptual bottleneck: visual hallucinations arise due to the models' inability to distinguish task-relevant objects from distractors. In principle, accurate identification and focus on critical objects while filtering out irrelevant ones is the key to break this limitation. A straightforward solution is one-step focus: directly attending to essential objects. However, this approach proves ineffective because effective focus inherently requires deep scene understanding. To this end, we propose *SceneDiver*, a coarse-to-fine focus plan generation method for VLMs leveraging their long-term planning abilities, that first constructs a holistic scene graph to establish initial comprehension, then progressively decomposes the task into simpler sub-problems through an iterative cycle of recognition, understanding, and analysis. To enable reactive control, we also design a lightweight adapter for distilling the deliberate focus ability into VLAs. Evaluations on standard embodied AI benchmarks confirm that our method substantially reduces visual hallucinations for both VLMs and VLAs, while preserving computational efficiency in tasks requiring fast execution. Our code and data are released at: https://future-item.github.io/SceneDiver.

[1]State Key Lab of CAD&CG, Zhejiang University, China [2]Baiont Quant, China [3]Zhejiang Key Laboratory of Intelligent Medical Decision Support, China. Correspondence to: Yao-Xiang Ding <dingyx.gm@gmail.com>.

*Proceedings of the 43rd International Conference on Machine Learning*, Seoul, South Korea. PMLR 306, 2026. Copyright 2026 by the author(s).

## 1. Introduction

Embodied vision-language decision making tasks, such as robotic manipulation and navigation, unify the challenges of visual perception, language understanding, as well as planning and action (Sapkota et al., 2025). Vision-Language and Vision-Language-Action Models (VLMs & VLAs) are powerful tools for embodied decision making with different benefits: VLMs are better as deliberate high-level planners (Ahn et al., 2022; Driess et al., 2023), whereas VLAs are better as end-to-end reactive executors (Brohan et al., 2023a; Kim et al., 2025; Black et al., 2026).

However, despite their distinct functional advantages, both of them encounter a common perceptual bottleneck, manifested in object hallucination, where non-existent objects are spuriously predicted, and perceptual errors including object omission, erroneous attribute binding, and inaccurate instance counting within the same category. In principle, accurate identification and focus on critical objects while filtering out irrelevant ones is the key to break this limitation. A straightforward solution is one-step focus: directly attending to essential objects with existing established visual focusing strategies. However, decision making in complex visual environments often demands progressive scene exploration to gradually disentangle task-relevant objects from cluttered backgrounds and verify their identity across multiple scales, making one-step focus strategies ineffective. For example, as shown in Figure 1, we present two failure cases of one-shot focus. It can be observed that without a comprehensive understanding of the scene, VLMs exhibit systematic mis-grounding and attention shifts.

To address this challenge, we propose *SceneDiver*, an effective method to break the perceptual limitation of VLMs in embodied vision-language decision making, leveraging their autonomous planning abilities. SceneDiver enables VLMs to autonomously generate the *focus plan*, which can help VLMs to focus only on task-relevant objects in each decision-making step. The central challenge lies in enabling VLMs to identify task-relevant objects from a cluttered scene containing numerous distractors. To address this, we introduce a coarse-to-fine, two-stage focus plan generation pipeline. At the first coarse-grained plan stage, we convert raw image data into structured graph representations by con-

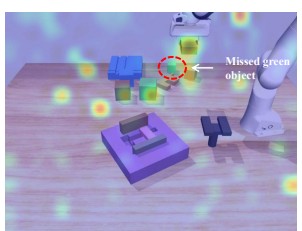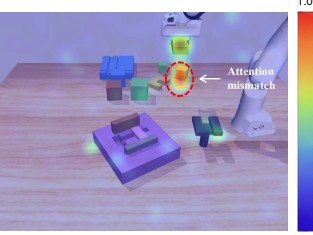

*Figure 1.* Failures in VLM decision making during direct text-image queries. Left: When queried about the number of green objects, the VLM's attention fails to capture the target objects, instead allocating excessive focus to irrelevant background elements. Right: When asked for the color of the object grasped by the robotic arm, the model exhibits attention mismatching; the focus incorrectly shifts to the yellow block rather than the target green object.

structing scene graphs. Graph reasoning is then performed over the constructed graph, decomposing the complex global scene into a series of simpler local sub-scenes corresponding to individual nodes in the graph. At the second fine-grained plan stage, VLMs are requested to autonomously explore each local sub-scene in agentic manner to discover task-relevant objects. Finally, the visual input is refined using the objects identified and fed to the VLM for the final decision. Essential task-related information is preserved, while irrelevant content is suppressed. Furthermore, to meet the latency requirements of real-time decision making, we design a lightweight adapter that distills our progressive focus planning into VLA models, allowing them to benefit from breaking perceptual ability while maintaining inference efficiency suitable for online deployment.

We conduct experiments across diverse tasks of robotic manipulation (Feng et al., 2025) and room navigation (Yang et al., 2025). The results demonstrate that our method achieves a 10%–15% performance gain in manipulation tasks and up to a 16% improvement in navigation, outperforming existing visual focus approaches. Then, we also evaluate our method on the LIBERO-plus benchmark (Fei et al., 2025) (which is employed to evaluate the robustness of the VLA against environmental interference). Experimental results demonstrate that our method achieves up to a 9.6% improvement in success rate, enhancing decision-making robustness while incurring a marginal computational overhead of only 2.64%.

## 2. Related Work

### 2.1. Vision-Language Decision Making

Vision-language decision making has become a promising research area (Ma et al., 2024). Benefiting from the perceptual capabilities of MLLMs and their comprehension of natural language instructions, traditional control policies have become more diversified and are now capable of interacting with users in a more intuitive manner. For instance, (Brohan et al., 2023b) employs MLLMs as high-level task planners, enabling the decomposition of user-provided instructions into a sequence of plausible low-level skills. Similarly, (Huang et al., 2022) adopts a two-stage framework to translate high-level instructions into executable actions. (Jang et al., 2022) achieves zero-shot task generalization to unseen tasks by aligning language instructions or human demonstration videos with visual observations of the environment. The performance of decision making in such systems is closely tied to the effectiveness of the vision encoder. Accordingly, numerous approaches have been proposed to enhance the visual processing components of MLLMs. (Shang et al., 2024) integrates knowledge distilled from diverse vision foundation models into a unified architecture to maximize visual representation capacity. (Assran et al., 2023) constructs an implicit world model by comparing embeddings of image patches to capture structural regularities. (Karamcheti et al., 2023) further improves vision-language alignment by incorporating language conditioning and generation into the masked image autoencoder (MAE) training objective.

### 2.2. Object Hallucination

Despite VLM's promising performance on various benchmarks (Danish et al., 2025; Li et al., 2025; Zong et al., 2024), in complex visual scenes, these models (Liu et al., 2023; Chen et al., 2024a) frequently perceive objects that do not exist in the provided images, a problem known as object hallucination (Rohrbach et al., 2019; Dai et al., 2023). Prior work has explored several avenues to mitigate this issue, including integrating an external object detector (Zhai et al., 2024), applying visually grounded visual instruction tuning (You et al., 2023; Zhang et al., 2024) or using reinforcement learning (Sun et al., 2023; Gunjal et al., 2024).

While prior work focuses on general object hallucination, the specific challenge of multi-object hallucination—where models invent multiple entities with incorrect attributes and relationships—remains less explored. Chen et al. (2025) finds that high cognitive loads from multi-object queries lead models to use heuristic shortcuts, bypassing rigorous visual analysis for each object. To mitigate this, researchers explore several avenues. Zhu et al. (2025) identifies that non-uniform spatial attention causes identical objects to be processed differently based on their position, and proposes a training-free attention rectification method to address this. Targeting a phenomenon termed "local overtrust", Huang et al. (2024) develops a method that monitors attention weights during decoding, which penalizes generation paths where a single token excessively influences subsequent content.

## 2.3. Scene Graphs

Scene graphs have been widely studied in vision-language research. Early work formulated scene graphs as structured representations of objects, attributes, and relations for semantic image retrieval (Johnson et al., 2015). Visual Genome (Krishna et al., 2017) later established scene graphs as a large-scale resource for dense scene understanding by providing annotations of objects, attributes, and relationships. Scene graphs have subsequently been used in visual question answering and reasoning, for example in GQA (Hudson & Manning, 2019), where they provide structured supervision and an explicit substrate for reasoning. Additionally, scene graphs have also been explored for image captioning, both as intermediate semantic representations for generation and as objects of analysis regarding their contribution to caption quality (Yang et al., 2019; Wang et al., 2019). Beyond vision-language tasks, scene-graph representations have also been extended to embodied and robotic settings. As representatives, Nguyen et al. (Nguyen et al., 2025) leverages 3D scene graphs as an intermediate representation of robot environments, which are converted into knowledge graphs to enable actionable reasoning for robot decision-making. Terra (Samuelson et al., 2025) constructs a hierarchical, terrain-aware 3D scene graph for task-agnostic outdoor mapping, underscoring the utility of structured scene representations in robotic reasoning. Finally, scene-graph representations have influenced evaluation as well: SPICE (Anderson et al., 2016) compares candidate and reference captions through semantic tuples derived from scene-graph-like structures rather than relying solely on n-gram overlap. Different from existing approaches, we let the scene graphs play the role as a structured prior for perception, helping break the perceptual bottleneck by enabling coarse-to-fine focus planning over task-relevant objects in cluttered scenes.

## 3. Method

### 3.1. Overview

Vision-Language Models (VLMs) frequently struggle with visual hallucinations and attention drift. While holistic scene understanding is essential and naive filtering risks discarding necessary information, processing the full, uncurated view exposes the model to task-irrelevant background clutter and spurious features that can mislead decision making. To address this, we introduce SceneDiver, a focus plan generation method that guides the model's attention through a coarse-to-fine reasoning process. In essence, we synthesize holistic graph-based reasoning with modulated perception to enhance decision making. Specifically, our system first executes high-level reasoning using a scene graph to identify critical regions of interest, denoted as *sub-scenes*. It then "looks closely" at these sub-scenes using a restricted percep-

tion field to iteratively verify their relevance. This process yields a pixel-level focus map that visually suppresses distracting content, effectively steering the model's attention toward essential targets. Finally, to facilitate real-time reactive control, we distill this explicit planning capability into a lightweight adapter for Vision-Language-Action (VLA) models.

### 3.2. Coarse Stage: Graph Reasoning

We utilize scene graphs as a structured prior to provide a coarse guide for the VLM's reasoning process. Using an OvSGTR (Chen et al., 2024b) backend, we generate a graph representation where nodes denote objects and edges encode spatial relationships. We input the scene graph information into the VLM in textual format. Specifically, detected objects are annotated with `<ref>`, and their spatial relationships are specified by `<pred>`. The location of each object is encoded using `<box>`. On this structure, the VLM executes high-level graph reasoning to identify task-relevant sub-scenes for subsequent processing. During this process, we require the VLM to produce concise, structured outputs at each step, using the graph's unique identifiers (`<ref>`) and spatial coordinates (`<box>`) to mark its regions of interest. We treat the VLM as the single source of truth and the graph as a reasoning guide for traversal. When inconsistencies arise, the VLM is free to select spatially proximate neighbors, discard if a target is irrelevant, or to retain ambiguous sub-scene as provisional candidates for the next stage.

### 3.3. Fine Stage: Verification and Exploration

The coarse graph reasoning provides a high-level route of sub-scenes, which is followed by a stage where the VLM sequentially zooms in on and processes each sub-scene. The model actively verifies whether information is relevant to the current task using a limited perception field, and autonomously selects, discards, or searches nearby areas to locate relevant information in an agentic manner. If the object within the local window aligns with the plan, it is immediately confirmed as a valid candidate, and the process moves to the next sub-scene. If the visual evidence is ambiguous, the model triggers a *semantic zoom*, narrowing its field of view to resolve fine-grained details. Conversely, if the expected target is absent due to graph-to-image misalignment, the model initiates a *local spatial search*, scanning the immediate neighborhood to attempt to re-acquire the target. This agentic behavior allows the system to recover from resolution limits or spatial drift without requiring hard-coded rules, naturally building the final set of verified objects $\mathcal{C}$ by solely utilizing the inherent capabilities of the VLM.

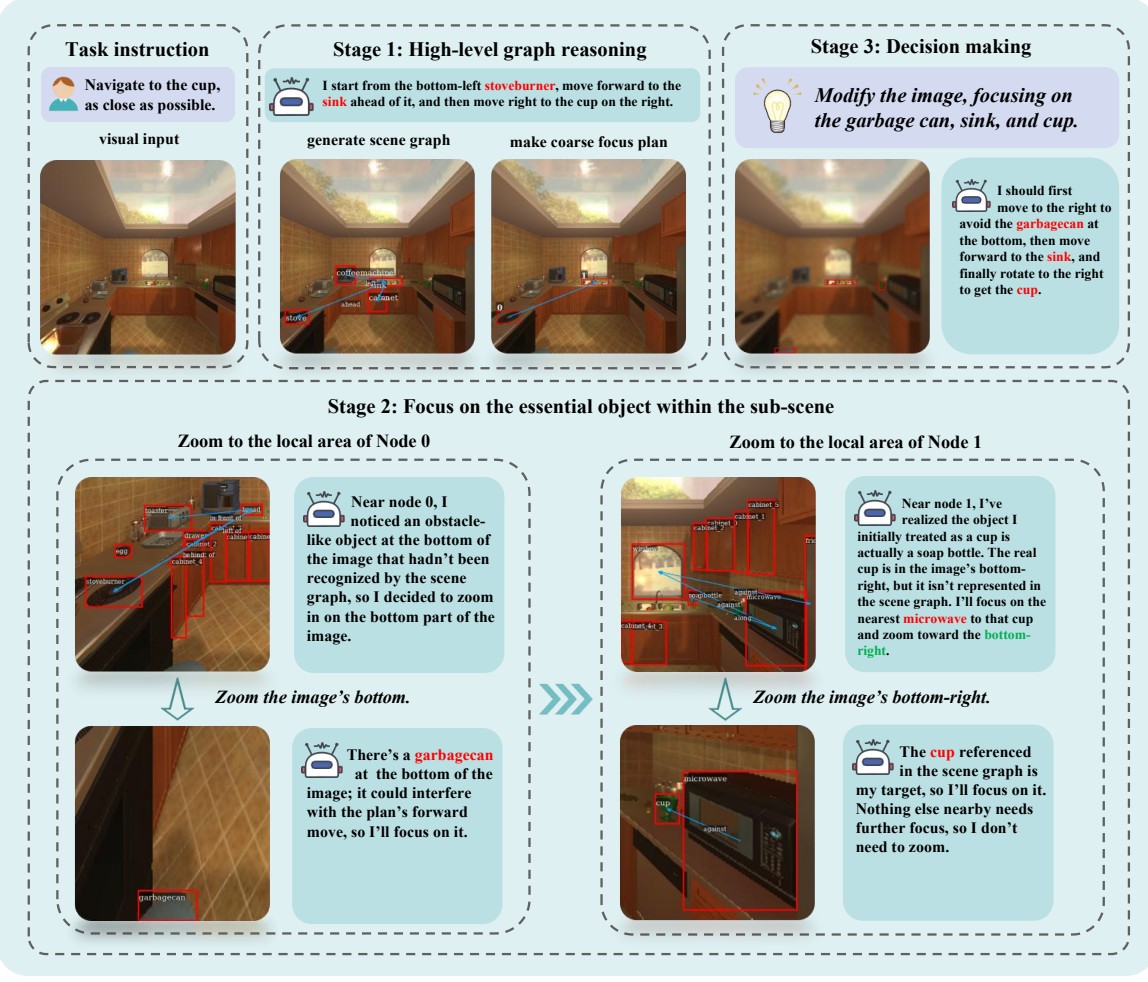

*Figure 2.* Overview of SceneDiver: From the input image we build a scene graph, perform graph reasoning to decompose the complex global scene into a series of simpler local sub-scenes corresponding to individual nodes in the graph (Stage 1), autonomously explore each local sub-scene using naturally designed exploration strategies to identify task-relevant objects (Stage 2), and use the resulting focus to modify the image and make a decision.

## 3.4. Guiding Decision through Focus Modulation

Upon establishing the candidate set $\mathcal{C}$ from the fine stage, we rasterize the candidate bounding boxes $\{b_k\}_{k \in \mathcal{C}}$ into a pixel-level Focus Score Map $s \in [0, 1]^{H \times W}$, defined as $s_{u,v} = \mathbb{I}[\exists k \in \mathcal{C} : (u, v) \in b_k]$. To direct attention without discarding environmental context, we empirically design a soft modulation that simultaneously attenuates luminance and high-frequency details in background regions based on the map. Given the input image $I$ and a visibility floor $\beta$ (to prevent total blackout), we first compute a luminance-scaled version $I_{\text{dim}}$. We then synthesize the final composite $I_{\text{out}}$ by blending this scaled image with its Gaussian-blurred counterpart $\mathcal{B}_\sigma$ inversely proportional to the score:

$$I_{\text{dim}} = I \odot \big(\beta + (1 - \beta)s\big), \tag{1}$$

$$I_{\text{out}} = s \odot I_{\text{dim}} + (1 - s) \odot \mathcal{B}_\sigma(I_{\text{dim}}). \tag{2}$$

This coupled operation keeps target objects sharp and bright while clutter is gently dimmed to reduce visual interference, fusing score map and raw image prompt into a modulated image prompt.

## 3.5. SceneDiver Adapter

Iterative scene graph traversal enables the VLM to make better decisions, while remaining computationally intensive. We further construct a real-time solution which we denote as SceneDiver adapter below, an end-to-end VLA adapter that distills explicit reasoning into implicit representations. Positioned after the projector in the cross-modal alignment space, it employs Slot Attention (Locatello et al., 2020) to extract task-conditioned structural representations, bridging visual tokens and logical entities. A mask prediction module then generates refined attention masks for task-relevant objects. Knowledge transfer is realized via two distilla-

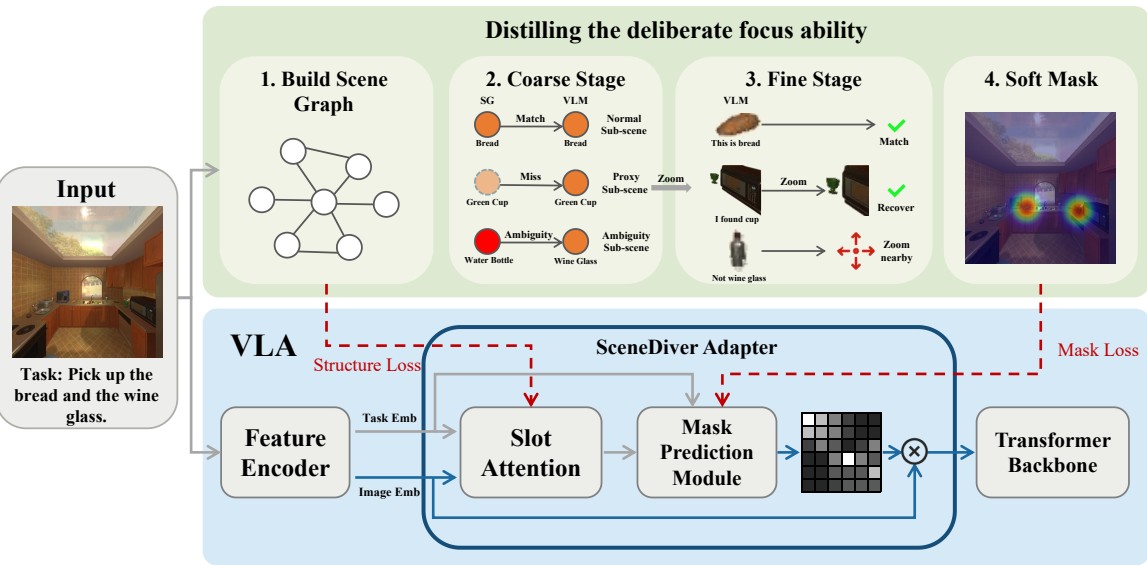

**Figure 3.** Distilling SceneDiver into a lightweight adapter to transfer the deliberate focus ability to a VLA for reactive control. Slot Attention learns the structured representations of scene graphs, while the mask prediction module learns the two-stage reasoning process.

tion objectives: a *Structure Loss* supervising Slot Attention to learn structured reasoning-related representations, and a *Mask Loss* supervising the mask prediction module to improve spatial localization accuracy.

**Task-Guided Slot Attention.** We project visual features $F \in \mathbb{R}^{L \times D}$ into object-centric slots $S \in \mathbb{R}^{K \times D_s}$, where $L$ denotes the number of visual patches, $K$ denotes the number of object-centric slots, and $D$ denotes the feature dimension of each patch (with $D_s$ the slot feature dimension). Unlike standard random initialization, we initialize this network by the following way: task tokens $T$ are pooled via learned attention into a global vector $v_{task}$. This vector conditions the initialization mean while variance remains a learnable global parameter:

$$S_{init} \sim \mathcal{N}(\mu(v_{task}) + \delta, \sigma_{global}). \quad (3)$$

This encourages slots to capture task-relevant entities rather than arbitrary textures.

**Scene-Aware Mask Prediction Module.** This module performs hierarchical reasoning analogous to the coarse-to-fine mechanism. At the **coarse stage**, it fuses slot semantics, slot mass (attention weight reflecting object scale), and task context to predict relevance scores $r_k$. At the **fine stage**, existing attention maps $A \in \mathbb{R}^{K \times L}$ propagate slot semantics back to patches, producing residual corrections:

$$M_{pred} = \sigma \left( \sum_k r_k \cdot A_{k,:} + \alpha \cdot \Delta_{patch} \right), \quad (4)$$

where $\alpha$ is initialized near zero, allowing the network to rely initially on slot-level predictions before progressively

incorporating spatial refinements; $\Delta_{patch}$ is a per-patch residual correction predicted at the fine stage, $\sigma(\cdot)$ is the sigmoid function, and $M_{pred}$ is the predicted patch-level mask probability.

**Distillation and Safety.** We train the adapter via Hungarian Matching (Kuhn, 1955) between slots and scene graph-derived objects, targeting two levels: slot-level alignment for correct scene decomposition, and mask-level refinement for spatially precise focus maps. For robust deployment, we introduce entropy-based dynamic gating. When uncertainty (measured over ambiguous patches) exceeds a threshold, the system bypasses masking and feeds raw observations to the VLA, allowing for graceful degradation in ambiguous scenarios.

## 4. Experiments

Our experimental evaluation is organized into four main phases. First, we present the qualitative operation of Scene-Diver and compare its performance against baselines in the context of robotic manipulation (Feng et al., 2025). Second, we evaluate our method on room navigation tasks (Yang et al., 2025), benchmarking it against existing focus approaches designed to mitigate visual hallucinations; Experimental results demonstrate that our method significantly outperforms other focus approaches in VLM-based decision making tasks. Third, we evaluate our method on LIBERO-Plus, a benchmark specifically designed for VLA robustness evaluation, using OpenVLA-OFT as the base model. Experimental results indicate that our method effectively bolsters the robustness of the base model while incurring

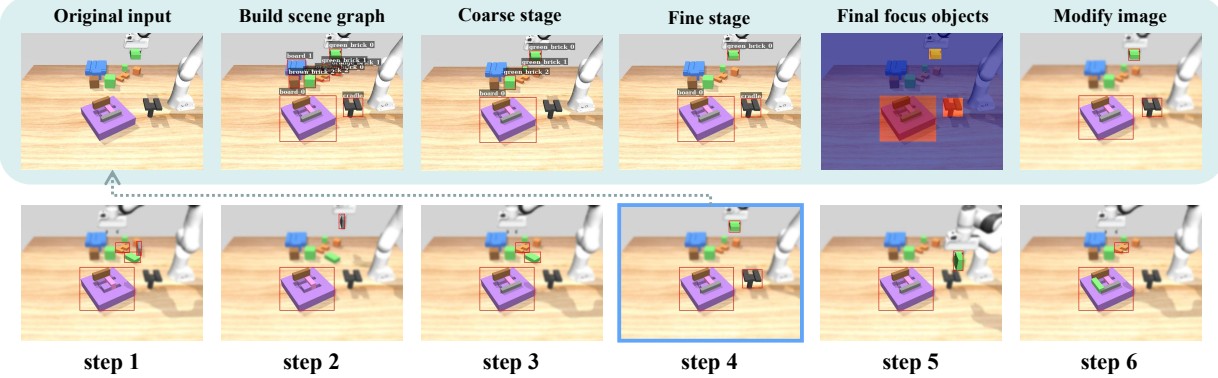

*Figure 4.* Qualitative visualization of the SceneDiver execution process for robot manipulation. The focus of each step is highlighted with a red bounding box (for visualization purposes only; the box is invisible to the VLM). The workflow of SceneDiver is illustrated using Step 4 as a representative example.

minimal computational overhead. Finally, we perform comprehensive ablation studies to validate the necessity of each algorithmic component and demonstrate the system's robustness against noisy or erroneous scene graphs through stress testing.

### 4.1. Robotic Manipulation

In the first experiment, we constructed a robotic-arm task in MuJoCo (Todorov et al., 2012) following the setup of Feng et al. (2025). The scene consists of a base plate with small pieces randomly scattered across a tabletop, augmented by a decoy board and several decoy pieces that serve as distractors. The agent's objective is to distinguish the target bricks from the distractors and assemble them onto the base plate. The control policy utilizes a discrete action space: *pick up*, *insert*, *reorient*, and *put down*. We randomly sampled 30 distinct scenes to formulate challenging experimental instances. In each instance, the VLM is provided with images of the current and target states and must complete the assembly within a strict horizon of 30 steps. The results are reported in Table 1, showing that our method generalizes effectively across multiple backends and significantly enhances decision accuracy in cluttered settings.

We illustrate a concrete decision trajectory in Figure 4. The red boxes are for visualization only and are not visible to the VLM. We present the process of SceneDiver at step 4. In the coarse stage, the VLM infers from the task and scene graph reasoning that the green block on the robotic arm may be the assembly target. However, since there are multiple green objects in the scene, it cannot determine whether the grasped object is the correct one. Therefore, in the fine stage, the VLM focuses on each green block individually, discovering that while the block on the gripper has the correct shape, its orientation is incorrect and requires rotation. Consequently, attention must also be paid to the cradle for reorienting the block. Ultimately, the VLM identifies that attention should

*Table 1.* Robot manipulation accuracy(Success Rate %). Means and Standard Errors (in the parentheses) over 5 seeds are reported.

| Model | Base | Focus |
|---|---|---|
| Qwen2.5-VL-7B-Instruct-AWQ (Bai et al., 2025) | $14.7_{(1.8)}$ | $28.7_{(1.6)}$ |
| Qwen2.5-VL-32B-Instruct-AWQ (Bai et al., 2025) | $21.3_{(1.2)}$ | $31.3_{(0.8)}$ |
| gpt-4o-mini (OpenAI, 2024) | $28.7_{(1.6)}$ | $34.0_{(1.2)}$ |
| gemini-2.5-flash (Comanici & et al., 2025) | $38.7_{(1.2)}$ | $46.7_{(1.0)}$ |

be directed to the green block on the robotic arm, the board, and the cradle.

### 4.2. Room Navigation

Building upon the room-navigation benchmark by Yang et al. (2025), we propose a more demanding variant designed to push the limits of decision making. We retain the original protocol where the agent navigates using only visual observations and environmental feedback, but introduce two key challenges. First, we select scenes with substantially higher environmental complexity. Second, we make target objects harder to identify by making them smaller, more frequently occluded, and easily confusable with visually similar distractors, further increasing perceptual ambiguity.

We evaluate our method across four distinct sub-tasks to test different capabilities: (1) Base: uses concise, direct instructions; (2) Common-Sense: presents instructions in natural, conversational language; (3) Complex-Instruction: requires identifying targets within lengthy, intricate prompts;

*Table 2.* Quantitative comparisons (Success Rate %) over 5 random seeds under the room navigation benchmark. **CS**: Common Sense, **CI**: Complex Instruction, **VA**: Visual Appearance.

| | Open-Source Models | | | | | | | | | | | |
| --- | --- | --- | --- | --- | --- | --- | --- | --- | --- | --- | --- | --- |
| | Qwen2.5-VL-7B-Instruct-AWQ (Bai et al., 2025) | | | | Qwen2.5-VL-32B-Instruct-AWQ (Bai et al., 2025) | | | | InternVL2.5-8B (Chen et al., 2024c) | | | |
| Method | Base | CS | CI | VA | Base | CS | CI | VA | Base | CS | CI | VA |
| Base Model | $32.7_{(1.5)}$ | $30.7_{(0.6)}$ | $32.0_{(1.2)}$ | $27.3_{(2.4)}$ | $49.3_{(1.1)}$ | $43.3_{(1.9)}$ | $46.7_{(1.3)}$ | $43.3_{(0.9)}$ | $29.3_{(1.1)}$ | $22.7_{(1.1)}$ | $24.0_{(1.1)}$ | $21.3_{(1.5)}$ |
| SoM | $30.0_{(0.9)}$ | $31.3_{(1.2)}$ | $31.3_{(1.8)}$ | $29.3_{(2.2)}$ | $49.3_{(1.1)}$ | $44.7_{(1.2)}$ | $48.0_{(0.7)}$ | $44.7_{(1.5)}$ | $28.0_{(1.5)}$ | $24.0_{(1.1)}$ | $23.3_{(0.9)}$ | $24.0_{(1.1)}$ |
| Multi-Res | $29.3_{(1.7)}$ | $32.7_{(0.6)}$ | $34.0_{(1.1)}$ | $29.3_{(1.1)}$ | $52.7_{(1.1)}$ | $46.7_{(1.6)}$ | $49.3_{(1.1)}$ | $45.3_{(2.0)}$ | $31.3_{(2.8)}$ | $20.7_{(0.6)}$ | $30.7_{(1.5)}$ | $22.7_{(1.1)}$ |
| VCD | $34.7_{(2.0)}$ | $32.0_{(2.2)}$ | $32.7_{(2.4)}$ | $33.3_{(2.1)}$ | $51.3_{(1.5)}$ | $46.0_{(1.5)}$ | $45.3_{(1.5)}$ | $46.0_{(1.1)}$ | $33.3_{(1.3)}$ | $24.7_{(2.0)}$ | $28.0_{(0.7)}$ | $23.3_{(0.9)}$ |
| **Ours** | $\mathbf{44.0}_{(1.1)}$ | $\mathbf{36.0}_{(0.6)}$ | $\mathbf{37.3}_{(0.6)}$ | $\mathbf{35.3}_{(0.7)}$ | $\mathbf{56.7}_{(1.3)}$ | $\mathbf{52.7}_{(1.1)}$ | $\mathbf{53.3}_{(0.9)}$ | $\mathbf{49.3}_{(1.1)}$ | $\mathbf{39.3}_{(0.7)}$ | $\mathbf{32.7}_{(0.6)}$ | $\mathbf{33.3}_{(1.1)}$ | $\mathbf{25.3}_{(0.7)}$ |

| | Closed-Source Models | | | | | | | | | | | |
| --- | --- | --- | --- | --- | --- | --- | --- | --- | --- | --- | --- | --- |
| | gpt-4o-mini (OpenAI, 2024) | | | | gemini-2.5-flash (Comanici & et al., 2025) | | | | doubao-seed-1.6-flash (Guo & et al., 2025) | | | |
| Method | Base | CS | CI | VA | Base | CS | CI | VA | Base | CS | CI | VA |
| Base Model | $43.3_{(1.6)}$ | $37.3_{(2.4)}$ | $38.7_{(1.5)}$ | $37.3_{(1.7)}$ | $68.0_{(1.1)}$ | $62.0_{(0.7)}$ | $60.0_{(0.9)}$ | $55.3_{(1.2)}$ | $59.3_{(1.1)}$ | $40.0_{(1.3)}$ | $38.0_{(2.2)}$ | $32.7_{(1.5)}$ |
| SoM | $41.3_{(0.7)}$ | $38.0_{(2.0)}$ | $36.0_{(1.1)}$ | $38.7_{(1.5)}$ | $69.3_{(1.1)}$ | $62.7_{(1.7)}$ | $60.7_{(2.4)}$ | $56.0_{(2.6)}$ | $60.7_{(2.6)}$ | $40.0_{(1.6)}$ | $40.7_{(1.1)}$ | $34.0_{(0.6)}$ |
| Multi-Res | $44.7_{(1.5)}$ | $38.7_{(1.2)}$ | $41.3_{(2.0)}$ | $43.3_{(1.9)}$ | $70.0_{(1.9)}$ | $60.7_{(1.5)}$ | $61.3_{(2.0)}$ | $58.0_{(2.9)}$ | $62.0_{(2.4)}$ | $41.3_{(2.0)}$ | $42.0_{(1.2)}$ | $37.3_{(2.4)}$ |
| Thinking | - | - | - | - | $70.0_{(1.3)}$ | $63.3_{(1.3)}$ | $63.3_{(1.9)}$ | $56.7_{(1.0)}$ | $60.7_{(2.2)}$ | $41.3_{(1.2)}$ | $43.3_{(1.0)}$ | $36.7_{(1.3)}$ |
| **Ours** | $\mathbf{50.0}_{(0.9)}$ | $\mathbf{53.3}_{(0.9)}$ | $\mathbf{49.3}_{(1.1)}$ | $\mathbf{49.3}_{(0.6)}$ | $\mathbf{74.7}_{(1.2)}$ | $\mathbf{65.3}_{(0.7)}$ | $\mathbf{66.0}_{(0.6)}$ | $\mathbf{62.0}_{(1.2)}$ | $\mathbf{66.0}_{(1.1)}$ | $\mathbf{48.0}_{(0.7)}$ | $\mathbf{48.7}_{(1.2)}$ | $\mathbf{44.7}_{(0.7)}$ |

and (4) Visual-Appearance: specifies targets indirectly via shape and color attributes rather than semantic names. We benchmark both open-source and closed-source models, conducting five independent trials per task. Table 2 reports the mean decision accuracy and standard error these runs.

As detailed in Table 2, we benchmark our approach against four representative paradigms. First, regarding visual prompting methods like SoM (Yang et al., 2023), while its SAM-based tagging effectively discretizes objects, the dense markers often introduce excessive visual noise in cluttered scenes, obstructing the VLM's reasoning; conversely, our two-stage filtering isolates relevant objects without occlusion. Second, following the multi-scale input paradigm established by Monkey (Li et al., 2024b), LLaVA (Li et al., 2024a), and Sphinx (Lin et al., 2023), we feed images at varying resolutions (global low-res and local high-res crops) into VLMs. This method enhances fine-grained details but loses spatial information; in contrast, our method leverages Scene Graphs to preserve spatial topology and restricts zooming to selective candidates to ensure efficiency. Third, distinct from decoding strategies like VCD (Leng et al., 2023) which mitigate language priors by modifying output logits but fail to address root perceptual insufficiencies, our method actively enhances input fidelity, ensuring decisions are grounded in high-resolution visual evidence rather than statistical corrections. Finally, even thinking-enhanced models such as doubao-seed-1.6-flash-thinking and gemini-2.5-flash-thinking underperform our method, underscoring that extended reasoning alone cannot remedy perceptual deficiencies.

### 4.3. End-to-End Control: Robustness on LIBERO-plus

Next, we evaluate the effectiveness of our adapter on real-time decision-making VLA. Specifically, we validate whether our proposed adapter can effectively distill the deliberate focus ability into the policy, thereby improving its resilience against environmental interference compared to the OpenVLA-OFT baseline. We conduct these evaluations on the challenging LIBERO-plus benchmark. Unlike standard benchmarks, LIBERO-plus serves as a rigorous stress test for policy robustness, introducing complex lighting variations, texture randomization, and adversarial distractors to the standard LIBERO suite.

We evaluate in five interference types: *Objects Layout*, which involves randomized object placements; *Camera Viewpoints*, which varies camera perspectives; *Robot Initial States*, which randomizes the initial configurations of the robotic arm; *Background Textures*, which applies randomized background textures; and *Light Conditions*, which introduces randomized lighting variations.

Following the LIBERO-plus evaluation protocol, we trained both OpenVLA-OFT and our method exclusively on the original, interference-free LIBERO dataset. We employed a joint training strategy using a data mixture from the *libero spatial*, *libero object*, *libero goal*, and *libero long* suites, and subsequently evaluated performance on each task individually. The quantitative results are presented in Table 3, revealing a consistent performance advantage of our SceneDiver-augmented policy over the OpenVLA-OFT baseline across all task suites. Most notably, the proposed method exhibits

*Table 3.* Performance comparison on LIBERO-plus benchmark suites. We report the mean success rate and standard error over 3 seeds.

| Task Suite | Objects Layout | Camera Viewpoints | Robot Initial States | Background Textures | Light Conditions |
|---|---|---|---|---|---|
| Spatial | **94.43**$_{\pm0.41\%}$ (↑2.04) 
 92.39$_{\pm0.27\%}$ | **54.35**$_{\pm0.99\%}$ (↑5.19) 
 49.16$_{\pm0.56\%}$ | **29.49**$_{\pm0.30\%}$ (↑9.58) 
 19.91$_{\pm0.74\%}$ | **87.41**$_{\pm0.40\%}$ (↑4.74) 
 82.67$_{\pm0.79\%}$ | **97.37**$_{\pm0.88\%}$ (↑3.51) 
 93.86$_{\pm0.18\%}$ |
| Object | **76.82**$_{\pm0.27\%}$ (↑0.81) 
 76.01$_{\pm0.27\%}$ | **64.56**$_{\pm0.86\%}$ (↑2.06) 
 62.50$_{\pm0.51\%}$ | **15.72**$_{\pm0.00\%}$ (↑1.36) 
 14.36$_{\pm0.27\%}$ | **90.61**$_{\pm1.53\%}$ (↑0.87) 
 89.74$_{\pm0.66\%}$ | **99.50**$_{\pm0.17\%}$ (↑0.51) 
 98.99$_{\pm0.67\%}$ |
| Goal | **52.53**$_{\pm0.38\%}$ (↑3.17) 
 49.36$_{\pm0.25\%}$ | **52.74**$_{\pm0.72\%}$ (↑1.72) 
 51.02$_{\pm0.43\%}$ | **18.85**$_{\pm0.00\%}$ (↑6.29) 
 12.56$_{\pm0.26\%}$ | **91.09**$_{\pm1.21\%}$ (↑9.31) 
 81.78$_{\pm0.40\%}$ | **97.35**$_{\pm0.76\%}$ (↑3.41) 
 93.94$_{\pm0.76\%}$ |
| Long | **74.52**$_{\pm0.56\%}$ (↑9.24) 
 65.28$_{\pm0.76\%}$ | **42.97**$_{\pm0.54\%}$ (↑2.84) 
 40.13$_{\pm0.40\%}$ | **37.73**$_{\pm0.00\%}$ (↑5.41) 
 32.32$_{\pm0.40\%}$ | **90.00**$_{\pm0.00\%}$ (↑8.93) 
 81.07$_{\pm0.00\%}$ | **92.42**$_{\pm0.00\%}$ (↑7.95) 
 84.47$_{\pm0.76\%}$ |

exceptional resilience in the Long-horizon task suite, yielding substantial improvements of 9.2% in Objects Layout and 8.0% in Light Conditions. Since long-horizon manipulation is particularly susceptible to error accumulation triggered by visual ambiguity, these significant margins indicate that our latent instantiation effectively stabilizes the policy's attention, preventing the "attention drift" often observed in the baseline when environmental variables fluctuate.

Beyond temporal stability, the method demonstrates robust generalization against visual domain shifts. Across the Spatial, Object, and Goal suites, we observe steady gains, up to 9.6%. This suggests that by explicitly modeling the scene structure in the latent space, our approach successfully disentangles semantic object information from irrelevant high-frequency pixel variations. Consequently, the policy maintains robust execution even when the visual appearance deviates significantly from the training distribution, effectively mitigating the structural and semantic ambiguities highlighted in our motivation.

### 4.4. Ablation and Stress testing

In this section, we conduct ablation studies and scene graph stress tests on the room navigation task.

**Ablation 1: Efficacy of Textual Scene Graphs.** To assess whether structured scene knowledge alone suffices, we supplied scene graphs directly as auxiliary text input for decision making. Results indicate that while scene graphs provide useful priors, excessive textual verbosity disperses attention from visual features and introduces noise, yielding only marginal improvements.

**Ablation 2: Step-by-Step vs. Direct Focusing.** To validate the necessity of our two-stage planning strategy, we compare it against a "Direct Focus" baseline that directly predicts targets given scene graphs without coarse-to-fine reasoning. Results indicate that this approach only focuses on the target object itself without considering the surrounding scene context, making it prone to confusion when similar-looking distractors are present.

**Ablation 3: Impact of Fine-Grained Granularity.** Com-

plementing our global planning, our method enforces fine-grained focus at each node. To assess the criticality of this component, we evaluated performance using only coarse-grained planning without fine-grained verification. Results reveal that coarse planning alone introduces hallucinated objects due to imprecise grounding, confirming that fine-grained verification is a critical correction mechanism.

**Scene Graph Stress Testing:** Given that the scene graph serves as the key part of our algorithm, we conducted stress tests specifically within the context of room navigation. Since our scene graph is domain-specific, large-scale node loss is rare in real-world inference; thus, we simulated such failures by manually dropping a subset of nodes to evaluate robustness and to examine whether the lost nodes can be recovered through fine-stage exploration.

As shown in Table 5, our method demonstrates consistently high recovery rates and minimal hallucination rates across varying levels of node sparsity. Specifically, the recovery rate remains above 96% across all drop ratios (10% to 50%), while the hallucination rate and miss rate are both maintained below 2%. These quantitative results indicate that even in the event of partial scene graph collapse, our two-stage verification mechanism ensures algorithmic stability. For detailed qualitative analysis and visualization of the recovery process, please refer to the Appendix.

## 5. Conclusion

In this work, we address a critical perception bottleneck in embodied vision-language decision making, manifested in object hallucination and perceptual errors. We propose SceneDiver, a coarse-to-fine focus plan generation method for VLMs. We break the perception bottleneck, by enabling VLMs to autonomously generate the focus plan, which can help VLMs to focus only on task-relevant objects in each decision-making step. To enable reactive control, we also design a lightweight adapter for distilling the deliberate focus ability into VLAs. Extensive experiments demonstrate that SceneDiver significantly reduces visual hallucinations for both VLMs and VLAs.

*Table 4.* Ablation results (Success Rate %) over 5 random seeds under the room navigation benchmark. **CS**: Common Sense, **CI**: Complex Instruction, **VA**: Visual Appearance.

| | Open-Source Models | | | | | | | | | | | |
|---|---|---|---|---|---|---|---|---|---|---|---|---|
| | Qwen2.5-VL-7B-Instruct-AWQ (Bai et al., 2025) | | | | Qwen2.5-VL-32B-Instruct-AWQ (Bai et al., 2025) | | | | InternVL2.5-8B (Chen et al., 2024c) | | | |
| Method | Base | CS | CI | VA | Base | CS | CI | VA | Base | CS | CI | VA |
| Base Model | $32.7_{(1.5)}$ | $30.7_{(0.6)}$ | $32.0_{(1.2)}$ | $27.3_{(2.4)}$ | $49.3_{(1.1)}$ | $43.3_{(1.9)}$ | $46.7_{(1.3)}$ | $43.3_{(0.9)}$ | $29.3_{(1.1)}$ | $22.7_{(1.1)}$ | $24.0_{(1.1)}$ | $21.3_{(1.5)}$ |
| Ablation1 | $30.7_{(1.1)}$ | $28.0_{(0.7)}$ | $34.7_{(1.8)}$ | $31.3_{(1.5)}$ | $50.7_{(0.6)}$ | $45.3_{(1.5)}$ | $49.3_{(1.7)}$ | $45.3_{(0.7)}$ | $31.3_{(0.7)}$ | $25.3_{(1.5)}$ | $26.0_{(1.5)}$ | $23.3_{(0.9)}$ |
| Ablation2 | $34.0_{(2.2)}$ | $31.3_{(0.7)}$ | $32.7_{(0.6)}$ | $30.0_{(0.0)}$ | $51.3_{(1.2)}$ | $46.7_{(0.9)}$ | $48.0_{(0.7)}$ | $45.3_{(0.7)}$ | $32.7_{(0.6)}$ | $27.3_{(1.1)}$ | $28.7_{(0.7)}$ | $22.7_{(1.1)}$ |
| Ablation3 | $37.3_{(0.6)}$ | $31.3_{(1.2)}$ | $34.7_{(0.7)}$ | $32.0_{(1.5)}$ | $52.0_{(0.7)}$ | $46.7_{(1.6)}$ | $48.7_{(1.8)}$ | $46.0_{(0.6)}$ | $37.3_{(0.6)}$ | $28.7_{(1.8)}$ | $27.3_{(1.5)}$ | $23.3_{(1.3)}$ |
| **Ours** | $\mathbf{44.0}_{(1.1)}$ | $\mathbf{36.0}_{(0.6)}$ | $\mathbf{37.3}_{(0.6)}$ | $\mathbf{35.3}_{(0.7)}$ | $\mathbf{56.7}_{(1.3)}$ | $\mathbf{52.7}_{(1.1)}$ | $\mathbf{53.3}_{(0.9)}$ | $\mathbf{49.3}_{(1.1)}$ | $\mathbf{39.3}_{(0.7)}$ | $\mathbf{32.7}_{(0.6)}$ | $\mathbf{33.3}_{(1.1)}$ | $\mathbf{25.3}_{(0.7)}$ |

| | Closed-Source Models | | | | | | | | | | | |
|---|---|---|---|---|---|---|---|---|---|---|---|---|
| | gpt-4o-mini (OpenAI, 2024) | | | | gemini-2.5-flash (Comanici & et al., 2025) | | | | doubao-seed-1.6-flash (Guo & et al., 2025) | | | |
| Method | Base | CS | CI | VA | Base | CS | CI | VA | Base | CS | CI | VA |
| Base Model | $43.3_{(1.6)}$ | $37.3_{(2.4)}$ | $38.7_{(1.5)}$ | $37.3_{(1.7)}$ | $68.0_{(1.1)}$ | $62.0_{(0.7)}$ | $60.0_{(0.9)}$ | $55.3_{(1.2)}$ | $59.3_{(1.1)}$ | $40.0_{(1.3)}$ | $38.0_{(2.2)}$ | $32.7_{(1.5)}$ |
| Ablation1 | $44.0_{(1.5)}$ | $38.7_{(2.0)}$ | $40.7_{(1.1)}$ | $39.3_{(1.1)}$ | $68.7_{(2.0)}$ | $62.7_{(0.6)}$ | $61.3_{(0.7)}$ | $56.0_{(1.1)}$ | $60.7_{(0.6)}$ | $41.3_{(0.7)}$ | $39.3_{(1.5)}$ | $36.0_{(1.5)}$ |
| Ablation2 | $44.0_{(1.5)}$ | $44.0_{(1.1)}$ | $44.7_{(0.7)}$ | $42.7_{(0.6)}$ | $70.7_{(0.6)}$ | $64.0_{(0.6)}$ | $63.3_{(1.9)}$ | $58.7_{(0.7)}$ | $62.7_{(1.1)}$ | $44.0_{(1.1)}$ | $41.3_{(1.8)}$ | $40.0_{(1.6)}$ |
| Ablation3 | $46.0_{(1.1)}$ | $47.3_{(0.6)}$ | $42.7_{(0.6)}$ | $44.0_{(1.1)}$ | $72.7_{(0.6)}$ | $63.3_{(0.9)}$ | $63.3_{(2.1)}$ | $57.3_{(1.7)}$ | $63.3_{(1.3)}$ | $46.0_{(1.1)}$ | $40.7_{(1.8)}$ | $40.7_{(1.5)}$ |
| **Ours** | $\mathbf{50.0}_{(0.9)}$ | $\mathbf{53.3}_{(0.9)}$ | $\mathbf{49.3}_{(1.1)}$ | $\mathbf{49.3}_{(0.6)}$ | $\mathbf{74.7}_{(1.2)}$ | $\mathbf{65.3}_{(0.7)}$ | $\mathbf{66.0}_{(0.6)}$ | $\mathbf{62.0}_{(1.2)}$ | $\mathbf{66.0}_{(1.1)}$ | $\mathbf{48.0}_{(0.7)}$ | $\mathbf{48.7}_{(1.2)}$ | $\mathbf{44.7}_{(0.7)}$ |

*Table 5.* Performance of SceneDiver under varying Drop Ratios (DR) across 500 scenes. We report the Recovery Rate (RR), VLM Hallucination Rate (HR), and Miss Rate (MR) to evaluate the system's robustness.

| DR | RR ($\uparrow$) | HR ($\downarrow$) | MR ($\downarrow$) |
|---|---|---|---|
| 10% | 96.78 | 1.52 | 1.70 |
| 20% | 98.01 | 1.23 | 0.76 |
| 30% | 97.16 | 1.89 | 0.95 |
| 50% | 97.69 | 1.02 | 1.29 |

**Limitations and future work**. While SceneDiver demonstrates strong performance across multiple benchmarks, several directions remain for future exploration. Although our stress tests show robustness to scene graph noise, the generation module could benefit from further optimization to handle extremely dynamic scenes with rapid object motion. Additionally, while our method advances in long-horizon tasks, exploring its integration with hierarchical planning frameworks could potentially yield additional benefits for more complex multi-stage manipulation scenarios.

## Acknowledgement

This work was supported by National Natural Science Foundation of China (U23A20311, 62206245). We would like to thank the anonymous reviewers for their constructive suggestions.

## Impact Statement

This paper presents work whose goal is to advance the field of Machine Learning. There are many potential societal consequences of our work, none of which we feel must be specifically highlighted here.

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

# A. Appendix

## A.1. Extended Ablation Studies

To evaluate the role of the adapter, we compare it with an end-to-end model that predicts the focus directly from the input image. The experimental setup and evaluation metrics are consistent with those in the main text. The results are as follows:

*Table 6.* Performance comparison on LIBERO-plus benchmark suites. In each cell, the bold black values represent our method, while the smaller grey values below represent the end-to-end baseline. We report the mean success rate and standard error over 3 seeds.

| Task Suite | Objects Layout | Camera Viewpoints | Robot Initial States | Background Textures | Light Conditions |
|---|---|---|---|---|---|
| spatial | **94.43**±0.41% (↑1.22) 
 93.21±0.27% | **54.35**±0.99% (↑2.81) 
 51.54±0.71% | **29.49**±0.30% (↑6.04) 
 23.45±0.15% | **87.41**±0.40% (↑1.00) 
 86.41±0.20% | **97.37**±0.88% (↑1.06) 
 96.31±0.53% |
| object | **76.82**±0.27% (↑0.41) 
 76.41±0.13% | **64.56**±0.86% (↑1.20) 
 63.36±0.68% | **15.72**±0.00% (↑0.82) 
 14.90±0.00% | **90.61**±1.53% (↑0.43) 
 90.18±0.66% | **99.50**±0.17% (↑0.17) 
 99.33±0.00% |
| goal | **52.53**±0.38% (↑2.02) 
 50.51±0.38% | **52.74**±0.72% (↑1.15) 
 51.59±0.43% | **18.85**±0.00% (↑4.71) 
 14.14±0.00% | **91.09**±1.21% (↑2.63) 
 88.46±0.61% | **97.35**±0.76% (↑0.76) 
 96.59±0.38% |
| long | **74.52**±0.56% (↑6.03) 
 68.49±0.94% | **42.97**±0.54% (↑1.75) 
 41.22±0.40% | **37.73**±0.00% (↑3.56) 
 34.17±0.40% | **90.00**±0.00% (↑3.03) 
 86.97±0.54% | **92.42**±0.00% (↑2.46) 
 89.96±0.57% |

Compared with the direct end-to-end baseline, our method performs better in all 20 LIBERO-plus settings, with larger gains under harder distribution shifts, including robot initial states, object layouts, and camera viewpoints.

## A.2. Qualitative Analysis of Fine-grained Verification

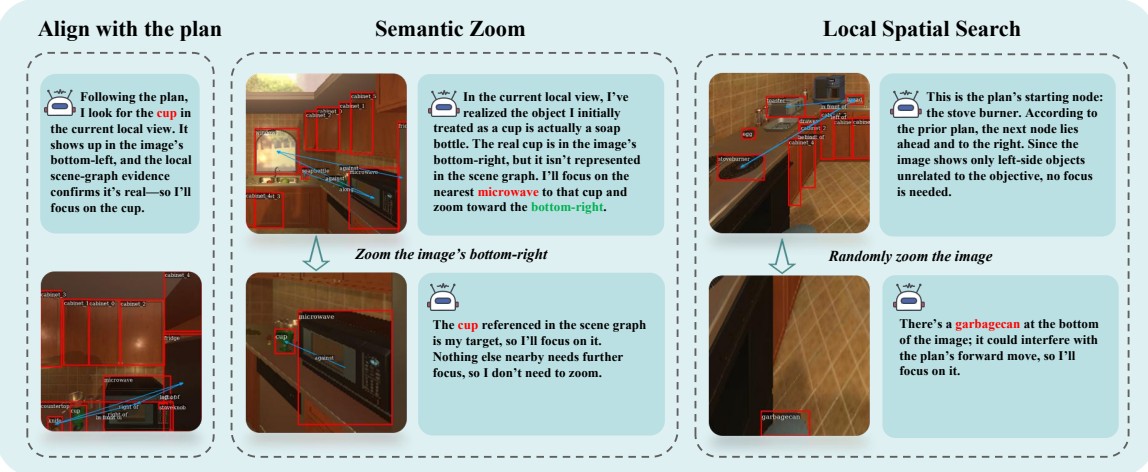

*Figure 5.* Fine-grained Verification and Exploration results of three different agentic behaviors

In the fine-grained exploration stage, we perform a zoom-in operation on each candidate node to construct a local scene graph, recovering details that may be invisible in the global view. Based on the local observations, the VLM adaptively executes one of three exploration primitives, as illustrated in Figure 5.

**Behavior 1.** When the target object is successfully detected within the local scene graph, the VLM performs multi-scale consistency verification to confirm that the object strictly meets the task requirements. This primitive encompasses two scenarios: (a) the target was identified in the coarse stage and is now re-confirmed locally, or (b) the coarse plan relied on a spatial proxy due to initial detection failures, and the zoom-in operation successfully reveals the originally missed target. Upon successful verification, the object is added to the final candidate set.

**Behavior 2.** When the target remains undetected after zoom-in—either because the local scene graph fails to capture it, or the detection results contradict the VLM's analysis—but a valid spatial sub-scene exists, the VLM leverages nearby objects

in the scene graph as proxies. Using their coordinates, the model performs a further zoom into specific regions for deeper inspection.

**Behavior 3.** When the target remains undetected and no clear sub-scene is available from the scene graph, the VLM resorts to the overall visual context. It executes a directional zoom based on spatial heuristics to explore the neighborhood and recover implicit targets that are unmapped by the scene graph.

### A.3. Perceptual Error Analysis

We quantitatively evaluated perception error rate in the navigation environment, including: missed present objects (Miss Rate), predicting nonexistent objects (False Positive Rate), wrong attribute binding (Bind Error Rate), incomplete same-class instance detection (Multi-Instance Error Rate), and missed task target (Target Miss Rate). For each sample, we determined whether each of these five error types occurred based on the consistency between the model output and the ground-truth scene annotations, and we report the proportion of each error type in both the original environment and the modified environment used in the main text. The results are shown in the table below:

*Table 7.* Results of perceptual error rates, 5 different types are reported in percentage (%).

| Qwen2.5-VL-7B-Instruct-AWQ | MR | FPR | BER | MIER | TMR |
|---|---|---|---|---|---|
| original env | 46.6 | 2.3 | 40.1 | 75.4 | 41.7 |
| modified env | 47.8 | 3.3 | 42.6 | 75.6 | 60.0 |
| **gemini-2.5-flash** | **MR** | **FPR** | **BER** | **MIER** | **TMR(%)** |
| original env | 23.8 | 4.6 | 57.8 | 77.0 | 22.6 |
| modified env | 31.9 | 2.9 | 53.3 | 74.6 | 42.9 |

### A.4. Experimental Results on the Vanilla Benchmark

In this section, we evaluate our method on vanilla EmbodiedBench and then explain why we also use Mujoco and modify the navigation task.

As shown in Table 8, our method still outperforms the base model on the vanilla navigation task. We choose modified navigation task because it introduces conditions that are more representative of real-world deployment, where perception is frequently challenged by occlusion, clutter, and partial observability.

*Table 8.* Result on Vanilla EmbodiedBench navigation task. Means and standard errors over 5 seeds are reported in percentage (%).

| Qwen2.5-VL-7B-Instruct-AWQ | base | common | complex | visual |
|---|---|---|---|---|
| base | $36.3_{(1.4)}$ | $25.0_{(0.5)}$ | $35.0_{(1.2)}$ | $28.3_{(1.2)}$ |
| ours | $40.7_{(1.6)}$ | $29.7_{(0.6)}$ | $38.0_{(0.8)}$ | $32.3_{(1.2)}$ |
| **gemini-2.5-flash** | **base** | **common** | **complex** | **visual** |
| base | $68.3_{(0.5)}$ | $65.0_{(0.5)}$ | $58.3_{(0.5)}$ | $56.7_{(0.5)}$ |
| ours | $71.7_{(0.5)}$ | $67.7_{(0.4)}$ | $61.0_{(0.4)}$ | $61.0_{(0.4)}$ |

We choose MuJoCo as the first experimental platform because it better matches the problem setting of this paper and offers stronger controllability. We focus on the perception bottleneck of VLMs under distracting conditions, and MuJoCo allows us to directly inject controllable distractions into the scene, making it better suited for validating the core mechanism of our method. In contrast, vanilla EmbodiedBench Manipulation does not fully align with the problem setting studied in this paper. As noted in EmbodiedBench (Yang et al., 2025), EB-Manipulation uses pre-given detection boxes and visual markers to help the model localize key objects. Such explicit visual cues substantially reduce the difficulty of perception, thereby weakening the perception bottleneck that this paper aims to investigate. To better match the problem setting studied in this paper, we removed the detection boxes while retaining only the axis information required for action execution. The experimental setup and evaluation metrics are consistent with those in the main text. As shown in the table below, our method still demonstrates a consistent improvement over the base model:

*Table 9.* Result on Vanilla EmbodiedBench manipulation task. Means and standard errors over 5 seeds are reported in percentage (%).

| Qwen2.5-VL-7B-Instruct-AWQ | base | common | complex | spatial | visual |
|---|---|---|---|---|---|
| base | $6.7_{(0.8)}$ | $8.8_{(0.8)}$ | $10.4_{(0.7)}$ | $15.0_{(0.8)}$ | $8.3_{(0.9)}$ |
| ours | $12.5_{(0.7)}$ | $10.4_{(0.7)}$ | $12.9_{(0.8)}$ | $16.3_{(0.8)}$ | $16.7_{(0.9)}$ |
| **gemini-2.5-flash** | **base** | **common** | **complex** | **spatial** | **visual** |
| base | $20.8_{(0.7)}$ | $14.6_{(0.7)}$ | $22.9_{(0.7)}$ | $24.2_{(0.5)}$ | $23.3_{(0.7)}$ |
| ours | $28.3_{(0.5)}$ | $17.9_{(0.5)}$ | $26.3_{(0.5)}$ | $28.3_{(0.5)}$ | $33.3_{(0.9)}$ |

## A.5. Inference Latency Breakdown of SceneDiver Components

*Table 10.* Detailed Latency Profiling and Tail Latency Analysis (ms)

| Module | Mean | Median | P95 | P99 | Percentage |
|---|---|---|---|---|---|
| Action (Total) | 114.45 | 113.81 | 116.50 | 121.78 | 100.00% |
| SlotAttention | 2.18 | 2.16 | 2.30 | 2.33 | 1.91% |
| MaskNet | 0.83 | 0.83 | 0.86 | 0.87 | 0.73% |

To evaluate the computational impact of the SceneDiver adapter on the OpenVLA-OFT model, we measured inference latency on a single NVIDIA RTX 4090 GPU. We benchmarked the latency of the Slot Attention and MaskNet modules over a sequence of 10,000 action generations. As shown in the table, the overhead introduced by SceneDiver (approximately 3.01ms) accounts for only 2.64% of the total action generation time (114.45ms). Crucially, the P99 latency of our modules remains within a very tight bound, ensuring the real-time reliability and determinism of the robotic system even in cluttered, multi-object environments.

## A.6. Implementation Details

### A.6.1. SCENEGRAPHGENERATION

We adopt the OvSGTR algorithm proposed by Chen et al. (2024b) as our primary scene-graph generator. Benefiting from its strong inherent generalization capabilities, the model achieves effective performance with a limited number of training samples. Our training dataset is synthesized from randomized scenes: for each scene, we extract 3D coordinates and semantic labels, then project these 3D points onto the 2D image plane using camera pose data. Finally, the model is trained using the official OvSGTR training protocols.

During the data generation process, each frame yields three synchronized outputs designed for scene-graph supervision:

- **Visual Data:** RGB images at a fixed resolution along with corresponding instance segmentation masks.

- **Object Annotations:** Visible objects are aligned with simulator metadata via unique identifiers. Category labels are normalized (lowercased with special characters removed), and compact 2D bounding boxes are computed from instance masks, filtering out any degenerate cases.

- **Relational Annotations:** OvSGTR/VG-style ⟨subject, predicate, object⟩ triples are generated to capture inter-object dependencies.

To support *geometry-aware* relations, our generator integrates image-space topology, depth ordering, and 3D metric geometry. The relational hierarchy includes an image-topology layer (e.g., *left of, right of, in front of, behind*) based on pixel-space proximity and median depth, and a geometric layer (e.g., *on, above, below, in*) determined by 3D physical overlap, vertical displacement, and containment links provided by the simulator.

This multi-scene, multi-view synthesis strategy, by combining complementary image-plane signals with metric 3D data, significantly enhances the robustness and generalization of the scene graph model against viewpoint variations, object articulations, and occlusions.

### A.6.2. SCENEDIVER ADAPTER

Our SceneDiver Adapter implements structured latent space reasoning through two core modules: Task-Guided Slot Attention and Scene-Aware MaskNet. The adapter is integrated between the visual projector and the LLM backbone of the VLA, operating directly within the vision-language alignment space.

**Task-Guided Slot Attention.** This module decomposes projected visual tokens (dimension $D = 4096$) into $K$ object-centric slot representations (dimension $d_s = 256$). Departing from the random initialization of standard Slot Attention, we utilize a Task-Conditional Initialization strategy. First, a global task vector $v_{task}$ is extracted from the task token sequence using a learned attention pooling mechanism:

$$w_i = \frac{\exp(s_i)}{\sum_j \exp(s_j)}, \quad v_{task} = \sum_i w_i \cdot h_i \tag{5}$$

where $s_i = \text{MLP}(t_i)$ represents the attention score for the $i$-th task token. The task vector is then projected to the slot dimension to serve as the mean for sampling, combined with learnable slot-specific offsets $\delta_k$ and a global variance $\sigma$:

$$S_0^k \sim \mathcal{N}(\text{Proj}(v_{task}) + \delta_k, \sigma) \tag{6}$$

Slots are refined through $T = 5$ iterations. In each iteration, we compute the slot-patch attention:

$$A_{kl} = \frac{\exp(\langle Q_k, K_l \rangle / \sqrt{d_s} \cdot \tau)}{\sum_{k'} \exp(\langle Q_{k'}, K_l \rangle / \sqrt{d_s} \cdot \tau)} \tag{7}$$

where $\tau = 0.4$ is a temperature parameter (below 1.0 to sharpen slot assignments and separate small objects). The weighted sum is fed into a GRUCell for state updates, followed by an MLP refinement.

**Scene-Aware MaskNet.** This module predicts the final focus mask based on refined slots using a coarse-to-fine approach:

**Slot-level Relevance Scoring (Coarse).** For each slot $s_k$, we fuse three signals: (i) slot semantic features $s_k$ (identity); (ii) **slot mass** $m_k = \sum_l A_{kl}/L$, projected via a 1→32D MLP to reflect scene coverage; and (iii) the global task context $v_{task}$. These are concatenated to predict a relevance logit $r_k \in \mathbb{R}$:

$$r_k = \text{MLP}([\text{LN}(s_k); \text{Proj}(m_k); \text{Proj}(v_{task})]) \tag{8}$$

**Patch-level Residual Refinement (Fine).** Slot semantics are back-propagated to the patch level using the attention map $A$ to build hybrid features $f_l = \sum_k A_{kl} \cdot s_k$. Combined with task context, an independent MLP predicts a local correction signal $\Delta_l$. The final patch logits are:

$$M_l = \sum_k r_k \cdot A_{kl} + \alpha \cdot \Delta_l \tag{9}$$

where $\alpha$ is initialized near 0 to ensure reliance on the slot-level baseline early in training. A learnable scale parameter $\exp(\beta)$ (initially $\beta = 1.1 \Rightarrow \text{scale} \approx 3$) is applied to slot logits to prevent ambiguous sigmoid outputs near 0.5.

### A.6.3. EXPERIMENT DETAILS

**Robot Manipulation.** To increase task difficulty, we introduce visual distractors in the scene. Specifically, we generate two types of distractor objects: (1) a fake assembly board with random grooves on its surface, placed in a separate region from the real task board, and (2) 3-5 small cubes with random sizes (6-10 voxels per side) and colors, scattered in off-board areas. These distractors share similar visual features with task-relevant objects but are not part of the assembly goal, requiring the model to distinguish between relevant and irrelevant objects.

**Room Navigation.** To select challenging target objects, we employ a distance-based sampling strategy using AI2-THOR scenes. For each floorplan, we filter small objects (volume $< 0.05$ m$^3$) that are visible and not structural elements (e.g., walls, floors). From these candidates, we either select the farthest object from the agent's initial position or randomly sample one that satisfies a minimum distance threshold. This ensures target objects are sufficiently challenging to locate and navigate to.

**Libero-plus.** For the Libero-plus benchmark, we use the same scene generation and task setup as the original LIBERO benchmark without introducing additional modifications.

### A.6.4. ADDITIONAL QUALITATIVE RESULTS

**Room Navigation.** In this section, we present detailed results for the navigation task across its subtasks. Each subtask uses a distinct instruction to probe different facets of the model's decision making. For every subtask, we provide two decision trajectories along with the corresponding instruction. For readability, we highlight the focused targets with red bounding boxes. Note that all annotations are invisible to the VLMs.

**Base.** In this subtask, the instruction is very explicit, for example: "navigate to the pot in the room and get as close as possible to it." The VLMs can directly observe the target object in the scene.

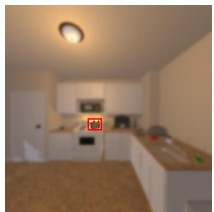 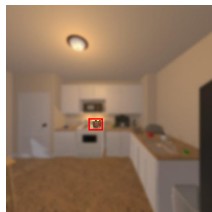 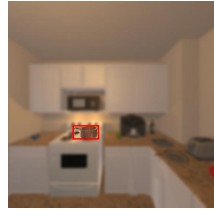 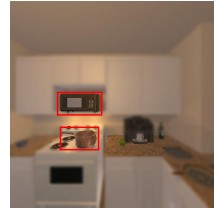 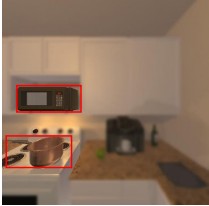

*Figure 6.* Instruction: navigate to the Pot in the room and be as close as possible to it.

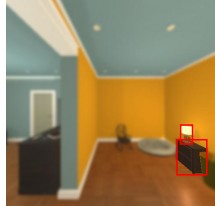 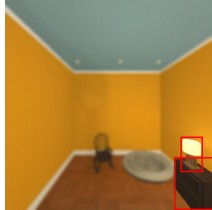 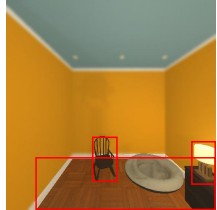 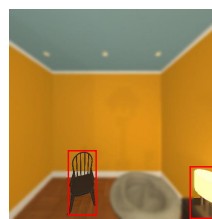 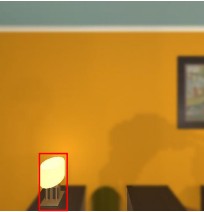

*Figure 7.* Instruction: navigate to the DeskLamp in the room and be as close as possible to it.

**Common Sense.** Compared to the base task, this task involves a more complex instruction, requiring stronger reasoning capabilities from the MLLMs.

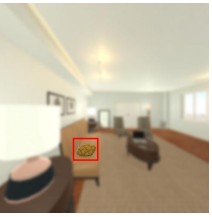 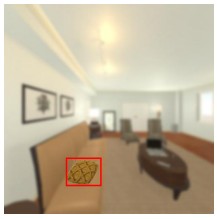 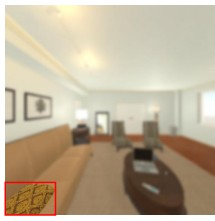 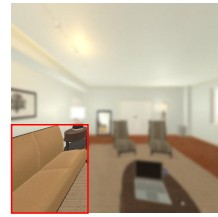 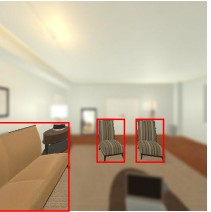

*Figure 8.* Instruction: I need a soft cushion to support my head while sleeping. Can you navigate to that object and stay close?

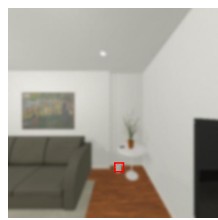 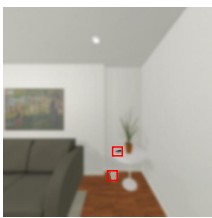 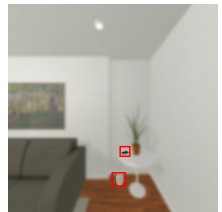 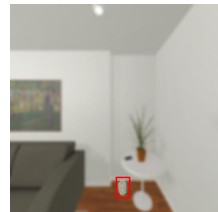 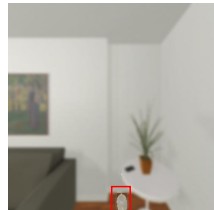

*Figure 9.* Instruction: I'd like to view a decorative sculpture representing a figure or person. Can you navigate to that object and stay close?

**Complex Instruction.** In this task, the instructions often contain substantial content irrelevant to the target, requiring VLMs to identify the task objective within complex prompts and align it with the objects in the image.

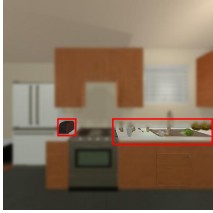 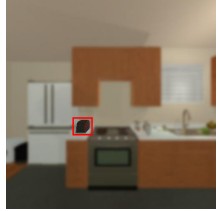 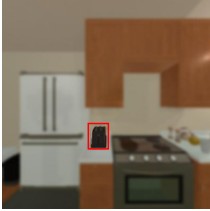 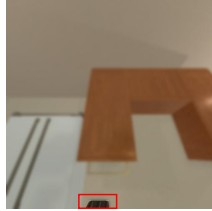 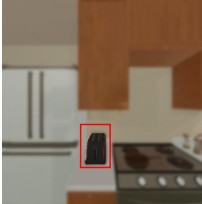

*Figure 10.* Instruction: The sound of someone walking upstairs adds a subtle rhythm to the quiet morning. There's a folded towel on the counter, and the air smells faintly of butter. Could you navigate to the toaster for me? It's a peaceful start to the day.

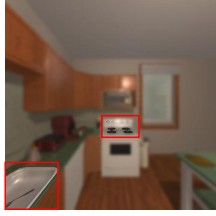 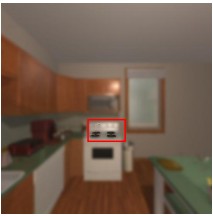 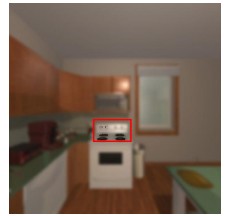 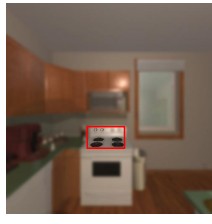 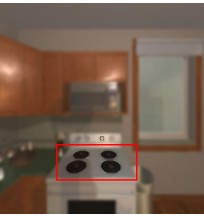

*Figure 11.* Instruction: The rhythmic ticking of the kitchen clock blends with the occasional drip from the faucet. There's a small pile of onions on the table, freshly chopped. Please move towards the stove burner for me. The kitchen has a comforting hum to it.

**Visual Appearance Task.** The instruction in this task describes the visual appearance of the target object, requiring the MLLMs to possess strong visual comprehension capabilities.

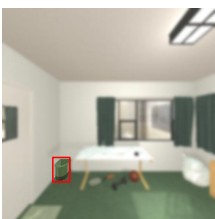 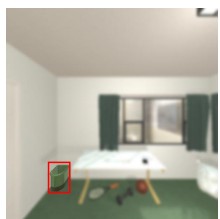 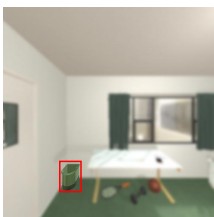 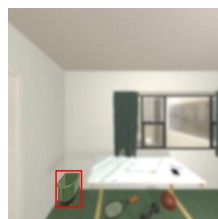 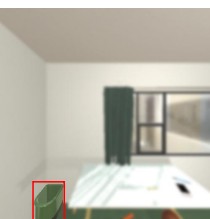

*Figure 12.* Instruction: Approach the tall green container with a smooth texture.

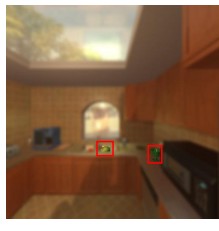 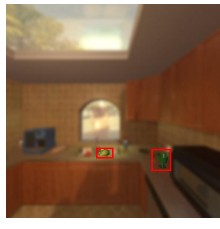 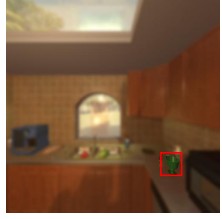 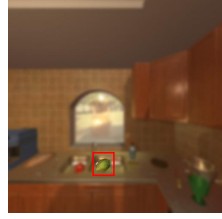 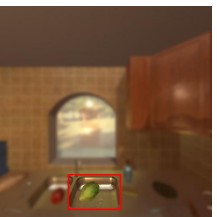

*Figure 13.* Instruction: Move closer to the small round object with a green surface and a cylindrical shape.

**Libero-plus.** In this section, we present the qualitative results of the SceneDiver Adapter on the LIBERO benchmark, illustrating the masking outcomes across various scenarios. Our method does not merely memorize objects from the training data; instead, it inherently learns to identify task-relevant entities and anticipate potential interactions during execution. This

demonstrates that the adapter has genuinely mastered the ability to focus on critical visual cues. Furthermore, our approach maintains a degree of generalization even in unseen configurations, such as camera shifts or robotic arm displacements.

Primary View

Wrist View

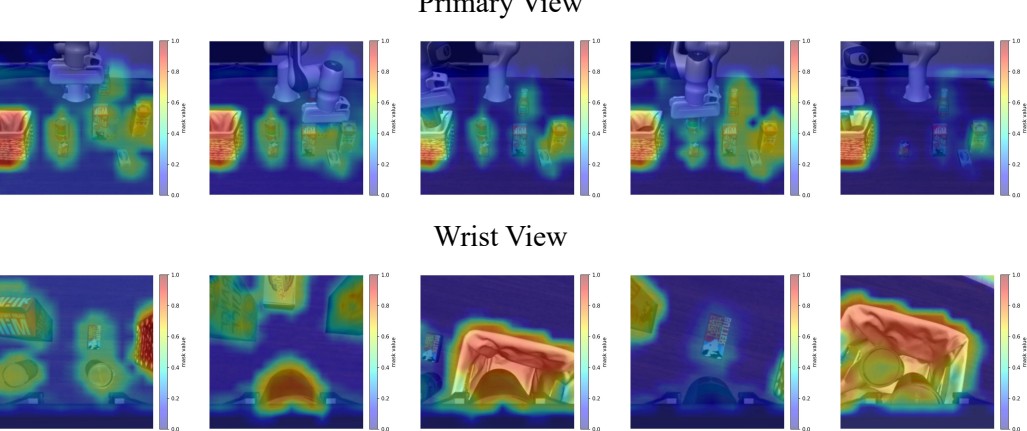

*Figure 14.* Instruction: Put both the alphabet soup and the tomato sauce in the basket.

Primary View

Wrist View

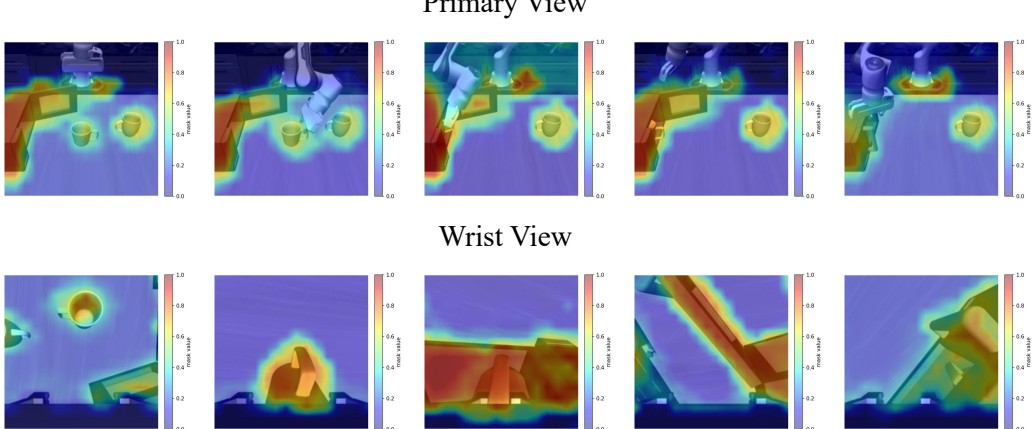

*Figure 15.* Instruction: Put the yellow and white mug in the microwave and close it.

Primary View

Wrist View

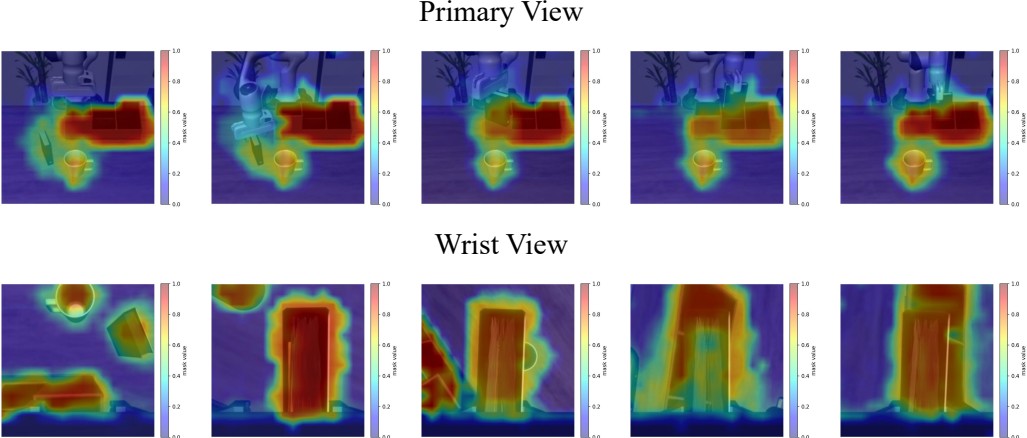

*Figure 16.* Instruction: Pick up the book and place it in the back compartment of the caddy.

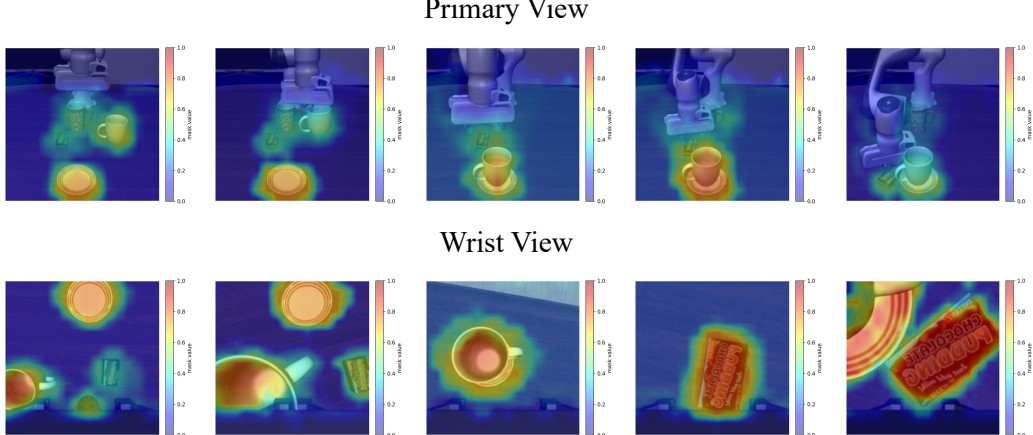

Primary View

Wrist View

*Figure 17.* Instruction: Put the white mug on the left plate and put the yellow and white mug on the right plate.

Primary View

Wrist View

*Figure 18.* Instruction: Put the white mug on the plate and put the chocolate pudding to the right of the plate.

