# OpenReview forum: "Dive into the Scene: Breaking the Perceptual Bottleneck in Vision-Language Decision Making via Focus Plan Generation"
_ICML.cc/2026/Conference — ICML 2026 regular_

### Official Review · Reviewer_Z4G2 · 2026-03-12

**Soundness:** 3
**Presentation:** 3
**Significance:** 2
**Originality:** 3
**Overall Recommendation:** 4
**Confidence:** 4

**Summary:**

This paper proposes a coarse-to-fine focus plan generation framework, SceneDiver, to solve visual hallucination and attention drift in embodied vision-language decision making. To achieve this, it introduces a two-stage reasoning pipeline including coarse scene-graph reasoning by decomposing the scene into sub-scenes, which can guide the model toward task-relevant regions before final decision making. It also proposes a lightweight adapter to distill this deliberate focus ability into VLAs, combined with a slot-attention-based mask prediction module. Extensive experiments on robotic manipulation, room navigation, and VLA robustness demonstrate improved decision accuracy and stronger robustness.

**Compliance With Llm Reviewing Policy:**

Affirmed.

**Key Questions For Authors:**

1. Section 4.4 sends the reader to Appendix A.1 for detailed results, but these ablations are central to justifying the method design and should not be placed entirely in the appendix. Could the authors move the key ablation table into the main paper?
2. "Progject" in Figure 3.
3. Could the authors provide more ablations for the remaining components, such as the adapter, structure loss, mask loss, and the uncertainty-based gating mechanism?

**Limitations:**

yes

**Strengths And Weaknesses:**

Strengths:
1. The paper is built on an intuitive motivation and presents a clear pipeline design. It starts from a concrete problem of attention drift and visual hallucination in cluttered embodied scenes, and then develops a focused solution around this problem. The proposed SceneDiver framework follows a coherent storyline. This makes the overall work easy to follow and gives the paper a strong sense of completeness.
2. The method is well integrated with existing VLM/VLA systems. Rather than replacing the backbone, the paper proposes a focus-planning mechanism that can be added on top of current models, and further distills this capability into a lightweight adapter for reactive control. This design makes the contribution more practical and potentially useful.
3. I appreciate the inclusion of robustness and scene graph stress testing analysis. The paper does not simply assume a perfect scene-graph extractor, but explicitly studies the effect of scene-graph corruption through node-dropping stress tests. This is a meaningful evaluation choice because the method depends heavily on structured perception. The reported results suggest that the fine-stage exploration can recover from partial graph failures while keeping hallucination and miss rates low, which supports the paper’s main claim.

Weaknesses:
1. I am not fully convinced by the problem formulation and the underlying assumption about attention drift. In embodied decision making, failure does not necessarily come from attending to too many regions, since VLMs often need broad scene-level context rather than object-detector-style localization. Some tasks require the model to jointly consider object relations, spatial layout, and surrounding context, so narrowing attention may not always be beneficial. In this sense, the paper sometimes seems to relate diffuse attention with erroneous attention too directly. Relatedly, the method introduces an additional scene-graph extraction stage, which itself can be noisy and may even be less reliable than the VLM backbone in some cases. This raises a broader question of whether the proposed pipeline is truly providing new capability, or whether it is mainly externalizing a reasoning process that a strong VLM could already perform through its own internal chain-of-thought style inference.
2. Part of the evaluation relies on custom benchmarks or modified settings whose difficulty is explicitly increased by the authors. This is not necessarily inappropriate, since the paper is targeting cluttered and distracting environments, but it makes it harder to judge how much of the improvement reflects genuine generalization and how much is tied to the specific failure mode that the method is designed to address. Under such a setting, gains are expected to favor a focus-oriented method. The empirical case would therefore be stronger if the paper also provided clearer results on the original benchmark without these modifications, or a more systematic analysis of performance as scene difficulty increases. In addition, the navigation experiments focus more on short-horizon target selection than on long-horizon planning, which limits the evidence for the claimed benefit of VLM reasoning.
3. I am concerned about efficiency and practical deployability. Latency is already a major challenge for VLA systems, especially in real-time embodied control, and the proposed framework adds several extra components, including scene-graph extraction, coarse-to-fine reasoning, and fine-stage verification. Even if each module is individually lightweight, the full pipeline may substantially increase end-to-end inference time. This is especially important because the paper also targets reactive control settings, where delayed decisions can directly reduce usefulness. As a result, I am not yet convinced that the method is practical for real-time deployment without a more detailed latency and efficiency analysis.

---

> ### Author Rebuttal · Authors · 2026-03-30
>
> We appreciate the reviewer’s valuable feedback and would like to respectfully clarify our position below.
>
> **Weakness**
> > 1\. I am not fully convinced by the problem formulation ...
>
> Our method combines global and local information in a two-stage process, rather than merely reducing the problem to local target localization. In the coarse stage, the model extracts global structural information, and on this basis, the fine stage narrows the visual scope for further verification and exploration. Through this two-stage information fusion process, our method helps the VLM better understand the scene, thereby breaking the perception bottleneck. Moreover, we are not externalizing the model’s internal reasoning, because when visual perception becomes a bottleneck, it is difficult to rely solely on the VLM’s own capability to extract complete decision-relevant information from dense visual inputs. Regarding the issue of scene graph noise, we have already validated in Section 4.4 of the main paper. As shown in Table 4, our method achieves over 96% accuracy in correcting the scene graph, demonstrating its robustness to errors in scene graph generation.
>
> > 2\. Modified settings, results on the original benchmark
>
> We would like to clarify two points:
> 1. Our method also improves performance on the original benchmark;
> 2. The modified environment is intended to mimic real-world clutter and distractors that induce perception bottlenecks.
>
> Using the same setup and metrics as in the main text, we evaluate on the original navigation benchmark, where our method still outperforms the base model.
>
> |Qwen2.5-VL-7B|base|common|complex|visual|
> |---|---:|---:|---:|---:|
> |base|36.3(1.4)|25.0(0.5)|35.0(1.2)|28.3(1.2)|
> |ours|40.7(1.6)|29.7(0.6)|38.0(0.8)|32.3(1.2)|
>
> |gemini-2.5-flash|base|common|complex|visual|
> |---|---:|---:|---:|---:|
> |base|68.3(0.5)|65.0(0.5)|58.3(0.5)|56.7(0.5)|
> |ours|71.7(0.5)|67.7(0.4)|61.0(0.4)|61.0(0.4)|
>
> We also compare perception-related errors between the original and modified environments to test whether the modified setting introduces new failure modes. We consider five error types: missed present objects (Miss Rate), predicting nonexistent objects (False Positive Rate), wrong attribute binding (Bind Error Rate), incomplete same-class instance detection (Multi-Instance Error Rate), and missed task target (Target Miss Rate). For each sample, we determine whether these five error types occur by comparing model outputs with ground-truth scene annotations, and report their proportions in both environments. The results are shown in the table below:
>
> |Qwen2.5-VL-7B|MR(%)|FPR(%)|BER(%)|MIER(%)|TMR(%)|
> |---|---:|---:|---:|---:|---:|
> |original env|46.6|2.3|40.1|75.4|41.7|
> |modified env|47.8|3.3|42.6|75.6|60.0|
>
> |gemini-2.5-flash|MR(%)|FPR(%)|BER(%)|MIER(%)|TMR(%)|
> |---|---:|---:|---:|---:|---:|
> |original env|23.8|4.6|57.8|77.0|22.6|
> |modified env|31.9|2.9|53.3|74.6|42.9|
>
> The results show that the modified setting introduces conditions that are more representative of real-world deployment, where perception is frequently challenged by occlusion, clutter, and partial observability.
>
> We will incorporate these two experiments into the revised manuscript.
>
> > 3\. Efficiency
>
> As detailed in Appendix A.3, the SceneDiver adapter adds only 3.01 ms overhead (2.64% of total action generation latency) while preserving stable tail latency, supporting its practicality for real-time deployment.
>
> **Question**
>
> > 1 and 2: move the ablation and "Progject"
>
> Thank you. We will revise accordingly.
>
> > 3\. More ablation
>
> To evaluate the role of the adapter, we compare it with an end-to-end model that predicts the focus directly from the input image. The experimental setup and evaluation metrics are consistent with those in the main text. The results are as follows:
>
> | Suite | Method | Objects Layout(%) | Camera Viewpoints(%) | Robot Initial States(%) | Background Textures(%) | Light Conditions(%) |
> | :--- | :--- | :--- | :--- | :--- | :--- | :--- |
> | spatial | Ablation | 93.21±0.27 | 51.54±0.71 | 23.45±0.15 | 86.41±0.20 | 96.31±0.53 |
> | | Ours | 94.43±0.41 | 54.35±0.99 | 29.49±0.30 | 87.41±0.40 | 97.37±0.88 |
> | object | Ablation | 76.41±0.13 | 63.36±0.68 | 14.90±0.00 | 90.18±0.66 | 99.33±0.00 |
> | | Ours | 76.82±0.27 | 64.56±0.86 | 15.72±0.00 | 90.61±1.53 | 99.50±0.17 |
> | goal | Ablation | 50.51±0.38 | 51.59±0.43 | 14.14±0.00 | 88.46±0.61 | 96.59±0.38 |
> | | Ours | 52.53±0.38 | 52.74±0.72 | 18.85±0.00 | 91.09±1.21 | 97.35±0.76 |
> | long | Ablation | 68.49±0.94 | 41.22±0.40 | 34.17±0.40 | 86.97±0.54 | 89.96±0.57 |
> | | Ours | 74.52±0.56 | 42.97±0.54 | 37.73±0.00 | 90.00±0.00 | 92.42±0.00 |
>
> Compared with the direct end-to-end baseline, our method performs better in all 20 LIBERO-plus settings, with larger gains under harder distribution shifts, including robot initial states, object layouts, and camera viewpoints.
>
> We will incorporate this experiment into the revised manuscript.

---

> > ### Author Rebuttal · Reviewer_Z4G2 · 2026-04-04
> >
> > Thanks for the reply. The new comparison with Gemini and the additional ablation study helps address my initial concern. I will maintain my score since I am still not convinced by the perspective from attention, but this does not influence the contributions of this paper.

---

> > > ### Author Response · Authors · 2026-04-06
> > >
> > > We sincerely thank you for your time and the thoughtful review. We appreciate your acknowledgment of the contributions of our work and fully respect your perspective on the attention-based interpretation.

---

### Official Review · Reviewer_yedU · 2026-03-16

**Soundness:** 3
**Presentation:** 3
**Significance:** 3
**Originality:** 2
**Overall Recommendation:** 4
**Confidence:** 4

**Summary:**

The  paper proposes SceneDiver, a coarse-to-fine method that uses scene graphs to guide VLMs to identify task relevant objects during embodied decision making (e.g. manipulation, navigation). The basic approach builds a scene graph, does graph-guided coarse reasoning to identify relevant sub-scenes, and then using VLM exploration to verify and refine focus at fine granularity. A lightweight adapter distils this into VLA models for lightweight real-time control. Evaluations are on a custom Mujoco manipulation setting, a modified EmbodiedBench nav split, and LIBERO-plus.

**Compliance With Llm Reviewing Policy:**

Affirmed.

**Final Justification:**

The rebuttal ultimately addressed my concerns.
I have no further concerns and still vote for weak accept.

**Key Questions For Authors:**

See above

**Limitations:**

yes

**Strengths And Weaknesses:**

Strengths:
- Well motivated problem with a relatively principled design. Coarse-to-fine makes sense. Dare I say it is intuitive.
- Excellent breadth of evaluation across manipulation, navigation and VLA, using both open and closed source models. Very thorough.
- Solid ablations. Table 5 shows that coarse-only, fine-only and direct focus all underperform the full pipeline. Convincing.


Weakness:
- I'm not convinced of the problem framing and motivation used in the paper, i.e. that visual hallucinations are the core problem that limits the capability of VLMs and VLAs, and that VLM attention maps focusing on the wrong objects (as in Figure 1) actually constitutes a hallucination. Identifying or counting the wrong object seems like a discrimination problem, not a hallucination problem (can't discriminate green vs. not green, or object vs. not-object). Further, attention maps are internal representations of the model, there are many different ways to compute them, and they don't always correlate with the models decisions. Hallucinations generally refer to models producing outputs that are not grounded in the inputs. If hallucination is now going to include discrimination failures and non-intuitive attention maps then the term hallucination has little meaning, may as well just say 'mistake'. Which is a long way to say that there is a method in the paper that is effective but framing it as a way of addressing hallucination is a bit of a stretch, particularly because hallucination rates are never directly measured. It's  possible (likely?) that the gains come from improved scene understanding rather than reduced hallucination per se.
- The related work is extremely narrow. There is all this previous work that used scene graphs (and coined the term) but none of it is mentioned. Consider for example Visual Genome (scene graph dataset, Kirshna, 2016), Image Retrieval using scene graphs (Johnson, 2015), scene graphs used for VQA, image captioning, evaluating image captioning (e.g. SPICE metric) and more. The paper could afford to spend a paragraph putting this in perspective.
- The paper evaluates on a modified version of EmbodiedBench with harder scenes and smaller and occluded objects. This make sense, but it wasn't clear to me how the baseline numbers were arrived - did the authors re-run the baseline models on the modified dataset?
- The MuJoCo evaluation is very small and not clearly described. It's not clear why EmbodiedBench manipulation tasks couldn't be used, or some other existing dataset.
- Is OvSGTR trained on synthetic data from the same simulators? (Appendix A.4.1). I'm not clear if there is any issue here that maybe we are overfitting to these simulators and would it generalize to other settings?

Overall:
- Despite a long list of weaknesses this is mostly me just trying to explain myself. It's my hope that all the questions can be explained, and minor issues addressed, which would enable the paper to be accepted.

---

> ### Author Rebuttal · Authors · 2026-03-30
>
> We thank the reviewer for the helpful feedback. We respectfully clarify our position below.
>
> **Weakness**
> > 1\. Problem framing, and quantitative analysis
>
> We quantitatively evaluated perception error rate in the navigation environment, including: missed present objects (Miss Rate), predicting nonexistent objects (False Positive Rate), wrong attribute binding (Bind Error Rate), incomplete same-class instance detection (Multi-Instance Error Rate), and missed task target (Target Miss Rate). For each sample, we determined whether each of these five error types occurred based on the consistency between the model output and the ground-truth scene annotations, and we report the proportion of each error type in both the original environment and the modified environment used in the main text. The results are shown in the table below:
>
> |Qwen2.5-VL-7B|MR(%)|FPR(%)|BER(%)|MIER(%)|TMR(%)|
> |---|---:|---:|---:|---:|---:|
> |original env|46.6|2.3|40.1|75.4|41.7|
> |modified env|47.8|3.3|42.6|75.6|60.0|
>
> |gemini-2.5-flash|MR(%)|FPR(%)|BER(%)|MIER(%)|TMR(%)|
> |---|---:|---:|---:|---:|---:|
> |original env|23.8|4.6|57.8|77.0|22.6|
> |modified env|31.9|2.9|53.3|74.6|42.9|
>
> The definition of hallucination in the original manuscript was not sufficiently clear, which may have caused misunderstanding. In this work, what we refer to as “hallucination” actually means the above five types of perception errors. We will provide this explicit definition in the revised manuscript to avoid conceptual ambiguity.
>
> We will incorporate this experiment into the revised manuscript.
>
> > 2\. Related work
>
> Thank you for pointing this out. We will include the relevant citations in the revised manuscript.
>
> > 3\. Re-run the baseline models?
>
> Yes, we reran all baselines on the modified dataset to fully assess their performance under our setting. The corresponding code and configurations will be released later to ensure reproducibility.
>
> > 4\. EmbodiedBench manipulation tasks and MuJoCo.
>
> EmbodiedBench Manipulation does not fully align with the problem setting studied in this paper. As noted in EmbodiedBench[1], EB-Manipulation uses pre-given detection boxes and visual markers to help the model localize key objects. Such explicit visual cues substantially reduce the difficulty of perception, thereby weakening the perception bottleneck that this paper aims to investigate. To better match the problem setting studied in this paper, we removed the detection boxes while retaining only the axis information required for action execution. The experimental setup and evaluation metrics are consistent with those in the main text. As shown in the table below, our method still demonstrates a consistent improvement over the base model:
>
> | Qwen2.5-VL-7B | base(%) | common(%) | complex(%) | spatial(%) | visual(%) |
> |---|---:|---:|---:|---:|---:|
> | base | 6.7(0.8) | 8.8(0.8) | 10.4(0.7) | 15.0(0.8) | 8.3(0.9) |
> | ours | 12.5(0.7) | 10.4(0.7) | 12.9(0.8) | 16.3(0.8) | 16.7(0.9) |
>
> | gemini-2.5-flash | base(%) | common(%) | complex(%) | spatial(%) | visual(%) |
> |---|---:|---:|---:|---:|---:|
> | base | 20.8(0.7) | 14.6(0.7) | 22.9(0.7) | 24.2(0.5) | 23.3(0.7) |
> | ours | 28.3(0.5) | 17.9(0.5) | 26.3(0.5) | 28.3(0.5) | 33.3(0.9) |
>
> It should be clarified that we chose MuJoCo as the experimental platform because it better matches the problem setting of this paper and offers stronger controllability. This paper focuses on the perception bottleneck of VLMs under distracting conditions, and MuJoCo allows us to directly inject controllable distractions into the scene, making it better suited for validating the core mechanism of our method.
>
> We will incorporate this experiment into the revised manuscript.
>
> > 5.1 Is OvSGTR trained on synthetic data from the same simulators?
>
> Yes.
>
> > 5.2 Generalize to other settings
>
> Thank you for the reviewer’s suggestion. We agree that extending the method to real-world scenarios is an important and valuable direction. However, existing related work has also mainly focused on simulator-only settings (e.g., EmbodiedBench [1] and LIBERO-Plus [2]), and we followed this setting when training the scene graph. The experimental results in simulation show that, for the problem defined in this paper, the performance of existing relevant methods is limited under this setting. In contrast, our proposed method achieves better results under the same setting, which verifies its effectiveness. We will consider generalization to real-world scenarios as an important direction for future work.
>
> [1] Yang R et al. EmbodiedBench: Comprehensive Benchmarking Multi-modal Large Language Models for Vision-Driven Embodied Agents. ICML 2025.
>
> [2] Fei et al. LIBERO-Plus: In-depth Robustness Analysis of Vision-Language-Action Models.

---

> > ### Author Rebuttal · Reviewer_yedU · 2026-04-05
> >
> > Thanks for the rebuttal. Some of my concerns are addressed.
> >
> > What does 'hallucination' mean? I think the rebuttal here has basically proved my point. The errors studied are: "missed present objects (Miss Rate), predicting nonexistent objects (False Positive Rate), wrong attribute binding (Bind Error Rate), incomplete same-class instance detection (Multi-Instance Error Rate), and missed task target (Target Miss Rate)". The only one of these that actually represents a hallucination is "predicted nonexistent objects" (1 of the 5 errors studied). Even then, the term hallucination is better suited to tasks that are more generative in nature. While the authors are free to come up with an entirely new definition of hallucination in the paper - that diverges from the generally accepted meaning - I just don't think it serves the reader to do so and can cause misunderstanding.
> >
> > Related work: I pointed out that the paper basically missed the entire history of the idea ('scene graph') that it is built upon - just saying "we'll fix it" in response with no draft is not very convincing.

---

> > > ### Author Response · Authors · 2026-04-06
> > >
> > > **1. Regarding the definition of hallucination:** We thank the reviewer for the further insightful discussions. To avoid conceptual ambiguity, we will adopt more precise terminology regarding hallucination throughout the revised manuscript. Specifically, we will use **object hallucination** for referring exclusively to "predicting nonexistent objects", whereas the five failure types studied in this work will be collectively defined as **perception errors**. Here, the term "perception errors" is used as an umbrella term for failures in visual scene understanding, similar to the usage in EmbodiedBench[1], where it describes failures arising at the visual state description stage.
> > >
> > > **2. Regarding the scene graph related work:** We will include a new subsection to discuss existing works about scene graph. The draft is as follows:
> > >
> > > > Scene graphs have been widely studied in vision-language research. Early work formulated scene graphs as structured representations of objects, attributes, and relations for semantic image retrieval[2]. Visual Genome[3] later established scene graphs as a large-scale resource for dense scene understanding by providing annotations of objects, attributes, and relationships. Scene graphs have subsequently been used in visual question answering and reasoning, for example in GQA[4], where they provide structured supervision and an explicit substrate for reasoning. Additionally, scene graphs have also been explored for image captioning, both as intermediate semantic representations for generation and as objects of analysis regarding their contribution to caption quality[5,6]. Beyond vision-language tasks, scene-graph representations have also been extended to embodied and robotic settings. As representatives, Nguyen et al. [7] leverages 3D scene graphs as an intermediate representation of robot environments, which are converted into knowledge graphs to enable actionable reasoning for robot decision-making. Terra[8] constructs a hierarchical, terrain-aware 3D scene graph for task-agnostic outdoor mapping, underscoring the utility of structured scene representations in robotic reasoning. Finally, scene-graph representations have influenced evaluation as well: SPICE[9] compares candidate and reference captions through semantic tuples derived from scene-graph-like structures rather than relying solely on n-gram overlap. Different from existing approaches, we let the scene graphs play the role as a structured prior for perception, helping break the perceptual bottleneck by enabling coarse-to-fine focus planning over task-relevant objects in cluttered scenes.
> > >
> > > We are truly grateful for your thoughtful comments and the time you devoted to reviewing our manuscript, which have helped us improve our work considerably.
> > >
> > > [1] Yang R et al. EmbodiedBench: Comprehensive Benchmarking Multi-modal Large Language Models for Vision-Driven Embodied Agents. ICML 2025.
> > >
> > > [2] Johnson et al. Image Retrieval using Scene Graphs. CVPR 2015.
> > >
> > > [3] Krishna et al. Visual Genome: Connecting Language and Vision Using Crowdsourced Dense Image Annotations. IJCV 2017.
> > >
> > > [4] Hudson D A, Manning C D. GQA: A New Dataset for Real-World Visual Reasoning and Compositional Question Answering. CVPR 2019.
> > >
> > > [5] Yang et al. Auto-Encoding Scene Graphs for Image Captioning. CVPR 2019.
> > >
> > > [6] Wang et al. On the Role of Scene Graphs in Image Captioning. In Proceedings of the Beyond Vision and LANguage: inTEgrating Real-world kNowledge (LANTERN) 2019.
> > >
> > > [7] Nguyen et al. Generating Actionable Robot Knowledge Bases by Combining 3D Scene Graphs with Robot Ontologies. IROS 2025.
> > >
> > > [8] Samuelson et al. Terra: Hierarchical Terrain-Aware 3D Scene Graph for Task-Agnostic Outdoor Mapping. Accepted to ICRA 2026.
> > >
> > > [9] Anderson et al. SPICE: Semantic Propositional Image Caption Evaluation. ECCV 2016.

---

### Official Review · Reviewer_jzZ5 · 2026-03-18

**Soundness:** 3
**Presentation:** 3
**Significance:** 2
**Originality:** 3
**Overall Recommendation:** 4
**Confidence:** 3

**Summary:**

This paper tackles the visual hallucination perceptual bottleneck in embodied vision-language decision-making for VLMs and VLAs, caused by models’ failure to distinguish task-relevant objects from distractors. The authors propose SceneDiver, a coarse-to-fine focus plan generation method that constructs scene graphs for holistic scene understanding and iteratively explores sub-scenes to locate critical objects. A lightweight adapter is further designed to distill the focus capability into VLAs for efficient reactive control. Experiments on robotic manipulation, room navigation and LIBERO-plus benchmarks show that SceneDiver significantly reduces visual hallucinations and improves task performance with minimal computational overhead.

**Compliance With Llm Reviewing Policy:**

Affirmed.

**Key Questions For Authors:**

1. How does the error of scene graph generation affect the final focus performance, and are there any lightweight fault-tolerant strategies?
2. Can the SceneDiver adapter maintain real-time performance in more complex long-horizon robotic tasks?
3. Have you tried to simplify the method and verify its generalization to general vision-language models?
4. Is there a quantitative metric to evaluate the agentic behaviors in the fine-grained exploration stage?

**Limitations:**

1. The method relies heavily on the quality of scene graph generation, and lacks effective optimization and verification for highly dynamic scenes with fast object changes.
2. The fine-grained exploration process completely depends on the inherent capability of VLMs, lacking explicit controllable mechanisms and sufficient interpretability.
3. It is only validated on embodied decision-making tasks such as robotic manipulation and navigation, without verifying its generalization to general vision-language models.
4. The real-time performance and robustness of the method in complex real-world robotic scenarios with severe interference have not been tested.

**Strengths And Weaknesses:**

Strengths
+ It targets the core perceptual bottleneck of visual hallucination in embodied vision-language decision-making. The proposed coarse-to-fine focusing method effectively solves the ineffectiveness of one-step focus, with a clear research motivation.
+ It integrates scene graph reasoning and iterative fine-grained verification, and designs a lightweight adapter for VLAs, balancing the long-term planning of VLMs and the real-time control of VLAs with a complete technical framework.
+ It is thoroughly evaluated on robotic manipulation, room navigation and LIBERO-plus benchmarks, achieving significant performance gains with negligible computational overhead.

Weaknesses
- The method is highly dependent on scene graph generation, and its robustness in highly dynamic scenes is not verified in depth.
- The agentic behaviors in fine-grained exploration rely entirely on VLMs without explicit controllable logic, resulting in poor interpretability.
- It is only validated on embodied decision-making tasks, without testing its generalization to general vision-language tasks.

---

> ### Author Rebuttal · Authors · 2026-03-30
>
> We thank the reviewer for the valuable feedback and clarify the main concerns below.
>
> **Weakness**
>
> > 1\. Highly dynamic scenes.
>
> Existing research on embodied decision-making is usually conducted in static or less dynamic environments (e.g., OpenVLA-OFT [1] and EmbodiedBench [2]). As shown in Section 4.4, our method already demonstrates robustness in such settings. Extending it to highly dynamic environments is an important direction for future research.
>
> > 2\. The agentic behaviors in fine-grained exploration.
>
> In fine-grained exploration, we employ three predefined strategies, and the VLM selects the most appropriate one based on the current situation, rather than determining the entire control logic from scratch. Instead of directly generating outputs with the VLM, our approach uses an explicit control structure for strategy selection. Ablation results show that, compared with directly relying on the VLM, this strategy-selection framework reduces errors caused by overlooking intermediate constraints and global context.
>
> > 3\. Generalization to general vision-language tasks.
>
> This paper focuses on embodied vision-language decision-making. More broadly, vision-language tasks also suffer from a perception bottleneck due to the need for scene understanding. The good performance of our method in embodied vision-language decision-making suggests that it may also be applicable to more general vision-language tasks, although exploring this possibility is beyond the scope of the present work.
>
> **Question**
> > 1\. How does the error of scene graph generation affect the final focus performance, and are there any lightweight fault-tolerant strategies?
>
> In Section 4.4, we experimentally analyze the impact of errors in scene graph generation on the final focus performance. As shown in Table 4, even when perturbations are applied to 50% of the nodes in the scene graph, our method still achieves an accuracy of over 96% in correcting the scene graph. These results demonstrate that our method is robust to errors in scene graph generation.
> In addition, our method incorporates two lightweight fault-tolerance strategies:
> 1. If the target instance is missing or has low confidence, we construct a temporary target from the instruction and context to continue exploration.
> 2. For inconsistent scene-graph names, such as different suffixed names for the same object, we resolve them via loose matching.
>
> > 2\. Real-time performance
>
> As shown in Appendix A.3, we measured the latency of 10,000 action generations on the LIBERO Plus dataset using a single NVIDIA RTX 4090 GPU. The dataset includes diverse environmental perturbations and long-horizon robotic manipulation tasks. Our results show that the SceneDiver adapter introduces only 3.01 ms of additional overhead, accounting for just 2.64% of the total action generation latency (114.45 ms). Moreover, the tail latency remains stable throughout the evaluation. These results indicate that the SceneDiver adapter can maintain real-time performance in complex long-horizon robotic tasks.
>
> > 3\. Generalization to general vision-language tasks.
>
> See weakness 3.
>
> > 4\. quantitative metric.
>
> In the fine-grained stage, the agent verifies the scene graph through exploration. Therefore, in Section 4.4 of the main paper, we quantitatively evaluate these agentic behaviors by perturbing the scene graph. As shown in Table 4, even when 50% of the nodes in the scene graph are perturbed, fine-grained exploration still corrects the scene graph with an accuracy of over 96%.
>
> **Limitation**
> >  1\. Highly dynamic scenes.
>
> See weakness 1.
>
> > 2\. The fine-grained exploration process.
>
> See weakness 2.
>
> > 3\. Generalization to general vision-language tasks.
>
> See weakness 3.
>
> > 4\. The real-time performance and robustness in complex real-world robotic scenarios.
>
> Thank you for the reviewer’s suggestion. We agree that extending the method to real-world scenarios is an important and valuable direction. However, existing studies have also mainly focused on simulator-only settings, such as EmbodiedBench [2] and LIBERO-Plus [3]. The experimental results in simulation show that, for the problem defined in this paper, the performance of existing relevant methods is limited under this setting. In contrast, our proposed method achieves better results under the same setting, which verifies its effectiveness. In addition, Appendix A.3 demonstrates that our method can satisfy real-time requirements. We will consider real-world validation as an important direction for future work.
>
> [1] Kim et al. Fine-Tuning Vision-Language-Action Models: Optimizing Speed and Success. RSS 2025.
>
> [2] Yang R et al. EmbodiedBench: Comprehensive Benchmarking Multi-modal Large Language Models for Vision-Driven Embodied Agents. ICML 2025.
>
> [3] Fei et al. LIBERO-Plus: In-depth Robustness Analysis of Vision-Language-Action Models.

---

> > ### Author Rebuttal · Reviewer_jzZ5 · 2026-04-05
> >
> > My concerns have been adequately addressed.

---

> > > ### Author Response · Authors · 2026-04-06
> > >
> > > We sincerely thank you for your time, constructive feedback, and valuable suggestions, which have greatly improved the quality of our manuscript.

---

### Decision · Program_Chairs · 2026-04-30

**Decision:**

Accept (regular)

**Comment:**

The submission introduces a coarse-to-fine focus plan generation method for VLMs leveraging their long-term planning abilities.  Reviewers raised concerns about its missing related work, incomplete experiments, and problem formulation.  The rebuttal was helpful, after which all reviewers turned positive, despite remaining concerns about its presentation.  The AC would like to follow the reviewers and recommend weak accept. The authors are encouraged to review the submission for the camera-ready.